# Co-option of epidermal cells enables touch sensing

**Federica Mangione** [1] ✉, **Joshua Titlow**[2], **Catherine Maclachlan**[3], **Michel Gho** [4], **Ilan Davis** [2], **Lucy Collinson** [3] **& Nicolas Tapon** [1] ✉

The epidermis is equipped with specialized mechanosensory organs that enable the detection of tactile stimuli. Here, by examining the differentiation of the tactile bristles, mechanosensory organs decorating the *Drosophila* adult epidermis, we show that neighbouring epidermal cells are essential for touch perception. Each mechanosensory bristle signals to the surrounding epidermis to co-opt a single epidermal cell, which we named the F-Cell. Once specified, the F-Cell adopts a specialized morphology to ensheath each bristle. Functional assays reveal that adult mechanosensory bristles require association with the epidermal F-Cell for touch sensing. Our findings underscore the importance of resident epidermal cells in the assembly of functional touch-sensitive organs.

Touch is an essential sensory modality through which animals gather information from the outside world by perceiving physical forces[1]. Air flows, gentle strokes and hair deflections are common tactile stimuli directly impinging on the outermost layer of animal bodies, the epidermis[1]. Detection of these stimuli relies on cutaneous mechanosensory organs, specialized cellular structures associated with mechanosensory neurons of the peripheral nervous system (PNS)[2–4]. Touch-evoked mechanotransduction, the conversion of tactile stimuli into neuronal impulses, occurs within cutaneous mechanosensory organs[5], yet still little is known about whether and how the surrounding epidermis sculpts the assembly of functional tactile organs. In this Article, to address this question, we used the mechanosensory bristles, tactile hairs decorating the adult *Drosophila* epidermis. Like mechanosensory hair follicle in the mammalian hairy skin[1], tactile bristles are the most abundant mechanosensory organs in the adult fly, decorating many body surfaces, including the dorsal cuticle overlaying the abdominal epidermis (Fig. 1a). To mediate mechanotransduction in response to touch, each bristle encloses a single mechanosensory neuron[6–8]. Each sensory bipolar neuron has its cell body and unbranched dendrite located in the periphery (Fig. 1a). Touch-evoked deflections of the hair shaft trigger electrical impulses, or action potentials, that propagate centrally along the ventral nerve cord[7,9].

Each mechanosensory bristle is composed of four lineage-related cells[6,10]: the epidermal shaft and socket cells, forming the tactile hair shaft and its base, and two subepidermal cells: the mechanosensory neuron itself and a glial-like sheath cell encapsulating the neuron dendrite and cell body. Adult abdominal bristles differentiate from a single sensory organ precursor (SOP) cell during pupal development. Between 14 and 24 h after puparium formation (hAPF), each SOP becomes committed to the neurogenic fate in response to Notch (N) signalling[11,12]. First, expression of neurogenic genes such as *neuralized* (*neur*) in the SOP, and expression of non-neurogenic genes in the surrounding epidermal cells downstream of N, specify tactile bristle precursor cells[13–15] (Extended Data Fig. 1a). The SOP then divides asymmetrically, in a N-dependent manner, to give rise to the epidermal socket and shaft cells and subepidermal neuron and sheath cells[6,16,17] (Extended Data Fig. 1b,c). Between 24 and 36 hAPF, the bristles initiate post-mitotic growth accompanied by the endoreplication of the socket and shaft cells, initiation of hair shaft outgrowth (Extended Data Fig. 1d,e), and the commitment to terminal differentiation as a self-contained mechanosensory organ[6,18–20].

## Results

### Differentiating bristles associate with epidermal F-Cells

To examine bristle differentiation, which remains poorly characterized, we used the bristle-specific *neur-GAL4* driver to express the nuclear reporter *H2B::RFP* (*neur > RFP*) and track cell behaviours over time (Fig. 1b). We noted that, while *neur > RFP* expression remained

[1]Apoptosis and Proliferation Control Laboratory, The Francis Crick Institute, London, UK. [2]Department of Biochemistry, University of Oxford, Oxford, UK. [3]Electron Microscopy Science Technology Platform, The Francis Crick Institute, London, UK. [4]Sorbonne Université, CNRS, Laboratoire de Biologie du Développement, Institut de Biologie Paris Seine (LBD-IBPS), Paris, France. ✉e-mail: federica.mangione@crick.ac.uk; nic.tapon@crick.ac.uk

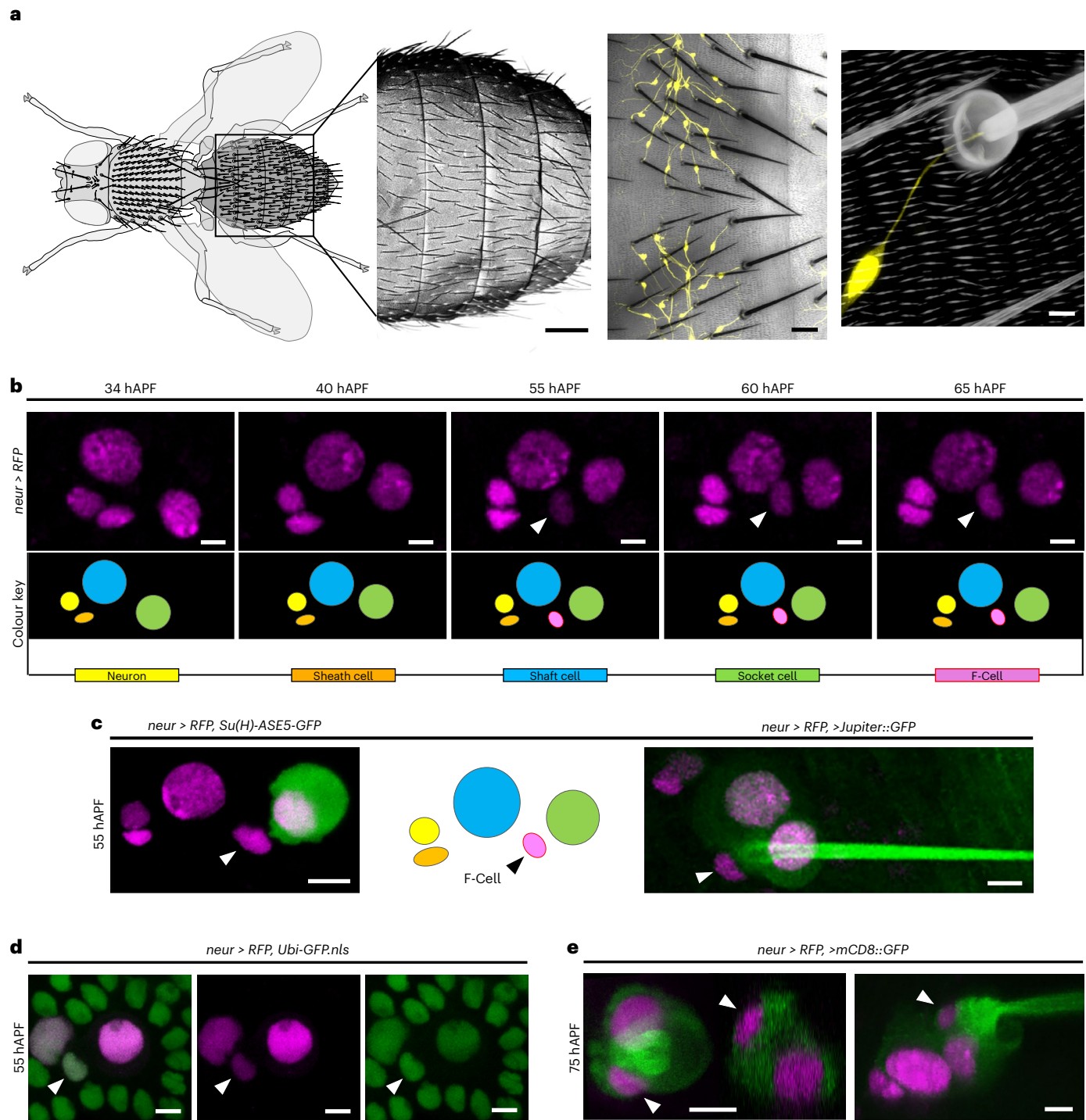

**Fig. 1 | Epidermal F-Cells associate with differentiating tactile bristles.**
**a**, Left to right: diagram of the adult fly highlighting the tactile bristles decorating the dorsal body surface, SEM image showing the hairy epidermal cuticle of the abdomen, bright-field image displaying the innervation of the tactile bristles, and close-up view of the cuticular socket and hair shaft structures showing the connection of the dendrite to the base of the tactile organ. PNS neurons are marked by *GFP* expression under *nSyb-GAL4* control (*nSyb > GFP*, yellow). **b**, Expression of the nuclear marker *H2B::RFP* under the control of *neur-GAL4* (*neur > RFP*, magenta) showing the time course of *neur* expression. Diagrams show each cell type as bristle differentiation progresses. By 55 hAPF, a fifth *neur > RFP* expressing cell, the F-Cell, is visible next to each organ (arrowhead). **c**, Left to right: tactile bristle and the F-Cell (arrowhead) marked by the expression

of *neur > RFP* (magenta) and the socket cell-specific reporter *Su(H)-ASE5-GFP* (green), diagram showing the position of the F-Cell relative to the bristle cells, and morphology of the tactile bristle and epidermis by co-expression of *neur > RFP* (magenta) and the ubiquitous microtubule marker *Jupiter::GFP* (green). Note that the F-Cell lies between the socket cell and the shaft cell. **d**, Simultaneous expression of *neur > RFP* with the ubiquitous nuclear marker *Ubi-GFP.nls* (green) shows that the F-Cell is part of the epidermis surrounding the bristle. **e**, Tactile organ visualized by co-expression of *neur > RFP* and the membrane-localized GFP *mCD8-GFP*, revealing association of the F-Cells (arrowhead) with the tactile bristles and the socket cell. Results are representative of three independent experiments. Scale bars, 150 µm, 20 µm, 5 µm (**a**) and 5 µm (**b**–**e**). Full genotypes are listed in Supplementary Table 1.

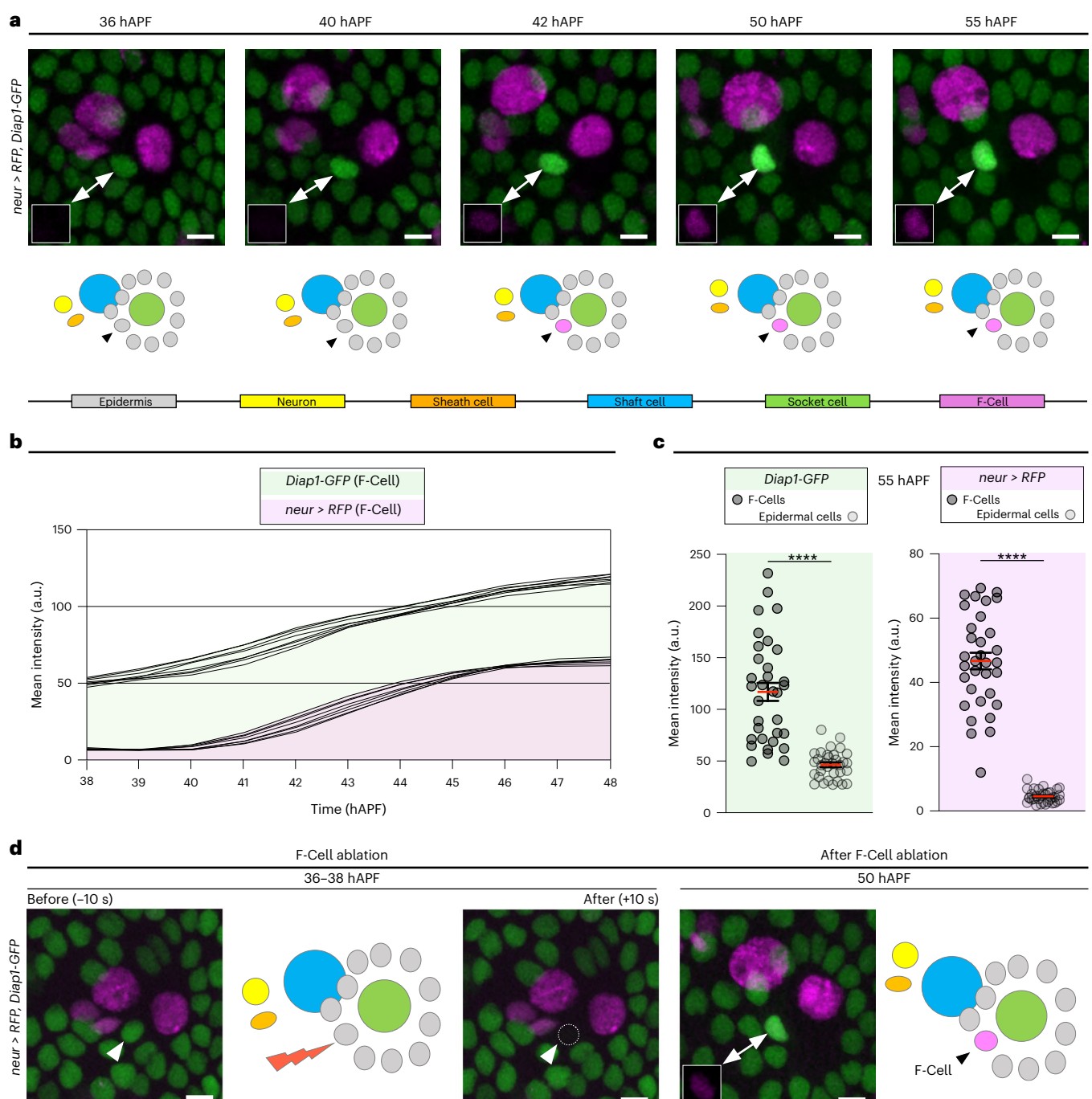

**Fig. 2 | F-Cell specification occurs post-mitotically. a**, Top: time-lapse imaging showing expression pattern dynamics of *neur > RFP* (magenta) and *Diap1-GFP* (green). The position of the epidermal F-Cell is indicated by arrowheads and insets show expression of *neur > RFP* in the F-Cell. Note that, over time, levels of *Diap1-GFP* are selectively upregulated in the F-Cell, allowing its unambiguous detection in the epidermal layer. Results are representative of three independent experiments. See Supplementary Video 1. Bottom: diagrams summarizing the post-mitotic specification of the F-Cell. **b**, Graph displaying changes in expression levels of *neur > RFP* and *Diap1-GFP* in the F-Cell between 38 hAPF and 48 hAPF (*n* = 8 bristles simultaneously imaged for RFP and GFP over the time course from three pupae and three independent experiments). **c**, Dot plots showing quantifications of *neur > RFP* and *Diap1-GFP* levels at 55 hAPF in epidermal F-Cells versus epidermal cells (*n* = 35 bristles simultaneously imaged for RFP and GFP over the time course from seven pupae and three independent experiments). Data are mean (red bar) ± s.e.m. (black bars). Two-tailed unpaired Student's *t*-test was performed; ****$P < 0.0001$. **d**, Elimination of the F-Cell via laser ablation at 36–38 hAPF leads to de novo specification of the F-Cell next to the tactile bristle and diagram showing the five *neur > RFP* expressing cells at 50 hAPF (*n* = 22 F-Cells over seven pupae from three independent experiments). Scale bars, 5 μm. Full genotypes are listed in Supplementary Table 1. Numerical data and exact *P* values are available in source data.

restricted to the bristle lineage between 36 hAPF and 48 hAPF, a fifth *neur > RFP*-expressing cell consistently appeared next to each differentiating bristle thereafter (Fig. 1b). Thus, as the bristles undergo

terminal differentiation, an additional fifth cell, hereafter called the F-Cell, begins to express the neurogenic marker *neur* (Fig. 1c). We next observed that F-Cells originate from the epidermis in which each

bristle is embedded. Labelling all cells by ubiquitous expression of nuclear GFP, together with *neur > RFP*, showed that F-Cells are initially indistinguishable from other epidermal cells that surround the bristle before they switch on the neurogenic marker *neur* (Fig. 1d). Notably, the F-Cell lies adjacent to the socket and shaft cells, as revealed by labelling *neur*-positive cells by simultaneous expression of nuclear *RFP* and a membrane-localized GFP (Fig. 1e and Extended Data Fig. 2a). Therefore, epidermal F-Cells become associated with the tactile bristles as these undergo terminal differentiation.

To distinguish the F-Cell from the other epidermal cells surrounding each bristle, we identified an enhancer of the *Death-associated inhibitor of apoptosis protein 1* gene fused with *GFP* (*Diap1-GFP*)[21] that showed dynamic expression in the F-Cell (Fig. 2a). We found that *Diap1-GFP* was expressed at equivalent levels in all epidermal cells surrounding the bristle, including the presumptive F-Cell, identifiable by its stereotyped position relative to the differentiating bristle by 36 hAPF (Fig. 2a). From 38 hAPF onward, *Diap1-GFP* levels increased specifically in the F-Cell, (Fig. 2a,b and Supplementary Video 1), making *Diap1-GFP* an unambiguous marker to identify F-Cells within the epidermis (Fig. 2c and Extended Data Fig. 2b,c). We also detected upregulation of *Diap1-GFP* in single epidermal cells next to the bristles of the thoracic epidermis (Extended Data Fig. 2d,e), suggesting that F-Cells associate with differentiating tactile bristles in different body parts.

### Bristles select F-Cells from the surrounding epidermis

The identification of the F-Cell provides a unique opportunity to examine how epidermal cells surrounding the bristles can acquire diversity during differentiation. Since upregulation of *Diap1-GFP* in the F-Cell occurs by 38–40 hAPF, well after the division of the bristle lineage cells (Extended Data Fig. 1b,c), we hypothesized that F-Cells might be specified post-mitotically. To test this hypothesis, we used a high-power femtosecond laser pulse to selectively ablate the F-Cell, then monitored the behaviour of the remaining epidermal cells (Extended Data Fig. 3a and Methods). Remarkably, ablating the F-Cell at 36–38 hAPF, before upregulation of *Diap1-GFP*, leads to de novo F-Cell specification (Fig. 2d and Extended Data Fig. 3b). Time-lapse imaging after F-Cell ablation revealed that a neighbouring epidermal takes over the position of the ablated cell, upregulates *Diap1-GFP* and switches on *neur > RFP* (Extended Data Fig. 3b,c and Supplementary Video 2). In contrast, a new F-Cell was not specified when ablations were performed after the onset of *neur > RFP* expression in this cell (Extended Data Fig. 3d,e). Therefore, F-Cell specification is a post-mitotic event and displays plasticity before the activation of *neur* expression.

The association of the F-Cell with the bristle, and its proximity to the socket and shaft cells (Fig. 1e and Extended Data Fig. 2), suggests that F-Cell specification relies on these cells. To investigate this possibility, we first performed paired ablations of the socket and shaft cells or of the neuron and sheath cells and examined F-Cell specification through *neur > RFP* and *Diap1-GFP* expression. While ablation of the subepidermal neuron and sheath cells had no effect (Extended Data Fig. 4a), simultaneous ablation of the non-neuronal shaft and socket

cells prevented F-Cell specification (Extended Data Fig. 4b). We next ablated the socket or shaft cells individually. We found that, in the presence of bristles lacking the socket cell, surrounding epidermal cells can still upregulate the expression of *Diap1-GFP* and express *neur > RFP*, albeit in an aberrant position (Extended Data Fig. 4c). Conversely, however, in the presence of bristles lacking the shaft cell, the upregulation of *Diap1-GFP* or expression of *neur > RFP* in epidermal cells was abolished (Extended Data Fig. 4d). Together, these data indicate that differentiating bristles recruit the F-Cell and suggest that the shaft cell is required for the initiation of F-Cell specification. To further explore this, we genetically altered the cell fate of the socket and shaft cells by modulating of N signalling[22,23]. N-activity gain in the shaft cell via downregulation of the transcriptional repressor Hairless (*H*), led to shaft-to-socket conversion[24,25], abrogating F-Cell specification (Extended Data Fig. 4e–g). Conversely, upon socket-to-shaft conversion[18,20] by upregulation of *H*, increased levels of *Diap1-GFP* were detectable in two cells surrounding the transformed organ made of two shaft cells (Extended Data Fig. 4f,g). These data further show that the shaft cell fate, established via N, is essential for the bristle to initiate F-Cell specification.

### EGFR signalling is required for F-Cell fate specification

Our data indicate that short-range signalling between differentiating bristles and surrounding epidermal cells is required for F-Cell specification. We focused on the epidermal growth factor receptor (EGFR) pathway since it is required for the specification of diverse cell types during adult PNS development, including the bract cells next to a subset of bristles of the legs and wings[26–28]. To visualize EGFR activity during F-Cell specification, we first visualized the expression of EGFR itself and observed higher levels in the F-Cell than epidermal cells (Extended Data Fig. 5a,b). We therefore monitored the expression of an *argos* (*aos*)-*GFP* reporter, which becomes upregulated in response to high EGFR activation[29,30] and found elevated *aos-GFP* expression in F-Cells (Fig. 3a and Extended Data Fig. 5c), supporting a requirement for EGFR signalling in F-Cell specification. To test this directly, we reduced EGFR activity by downregulating the expression of the EGF ligand *spitz* (*spi/EGF*) or downstream effector *rolled* (*rl*), which encodes ERK/MAPK[27,31–33]. In both conditions, *neur > RFP* and *aos-GFP* were no longer expressed in any epidermal cells surrounding the bristle (Fig. 3b and Extended Data Fig. 6a), indicating the lack of F-Cell specification. Impaired F-Cell specification was also revealed by the lack of *Diap1-GFP* upregulation and *neur > RFP* expression in the epidermal cells surrounding bristles upon downregulation of *spi/EGF*, expression of a dominant negative form of EGFR (*EGFR^DN*) or reduced *rl/ERK* activity (Fig. 3c–e and Extended Data Fig. 6b,c). As terminal differentiation of the bristle cells appeared unperturbed by EGFR manipulation (Extended Data Fig. 6d), these data indicate that activation of EGFR signalling is required for F-Cell specification.

The shaft cell and expression of *spi/EGF* in the bristle are required for F-Cell specification (Fig. 3c,d and Extended Data Fig. 4). We therefore hypothesized that the shaft cell can act as the source of Spi/EGF. We

**Fig. 3 | F-Cell specification requires EGFR signalling. a,** Expression pattern of the *aos-GFP* reporter (green) and *neur > RFP* (magenta) at 55 hAPF, showing high EGFR activity in the F-Cell. **b,** Images showing the lack of F-Cell specification upon downregulation of the EGFR ligand *spi/EGF* within the bristle through the absence of *aos-GFP* and *neur > RFP* in epidermal cells surrounding the bristle. **c,** Epidermal cells (*Diap1-GFP*, green) and bristle (*neur > RFP*), showing the F-Cell (arrowhead) at 55 hAPF. **d,** Images showing the lack of F-Cell specification upon downregulation of *spi/EGF* by uniform *Diap1-GFP* expression and absence of *neur > RFP* in the epidermal cells surrounding the bristles. **e,** Dot plots showing quantifications of *neur > RFP* (left) and *Diap1-GFP* (right) intensity fold changes in presence (*n* = 35 control bristles from seven pupae) or absence of EGFR signalling (*n* = 37 *spi/EGF^KD* bristles from five pupae; *n* = 25 *EGFR^DN* bristles from five pupae; *n* = 37 *rl/ERK* bristles from five pupae), each from three independent experiments. Note that

levels of both markers remain basal when EGFR signalling is off. Data are mean (red bar) ± s.e.m. (black bars). The two-tailed unpaired Kolmogorov–Smirnov test was performed; ****$P < 0.0001$. **f,g,** Expression of *Diap1-GFP* in the F-Cell (arrowhead) after expression of *EGFR^DN* in clones of cells within the epidermis (**f**) or in clones of cells including epidermal cells and the F-Cell (**g**). Clones of cells expressing *EGFR^DN* are marked in magenta (Methods). **h,** Morphology of the tactile bristle (left) and false coloured image (right) highlighting the association of the F-Cell (magenta) with the socket and shaft cells at 58 hAPF. **i,** Morphology of the tactile bristle and false coloured image highlighting the lack of associated F-Cell with the socket and shaft cells when EGFR signalling is downregulated in the bristle. Results are representative of three independent experiments. Scale bars, 5 μm. Full genotypes are listed in Supplementary Table 1. Numerical data and exact *P* values are available in source data.

directly tested this hypothesis by silencing *spi/EGF* expression exclusively in shaft cells (Methods), and found that this was indeed sufficient to abrogate F-Cell specification (Extended Data Fig. 6e,f). We next asked whether lack of EGFR in the presumptive F-Cell was sufficient to impair F-Cell specification. To address this, we co-expressed EGFR[DN] and membrane-localized *mCherry* stochastically in the epidermal cells during F-Cell specification (Methods) and found that, upon loss of EGFR activity in the F-Cell, the F-Cell fate was abolished (Fig. 3f,g). Accordingly, upon loss of EGFR signalling, differentiating bristles remained composed of only four cells (Fig. 3h,i). Together, these data show that Spi/EGF secreted by the shaft cell is required to specify F-Cell fate via EGFR.

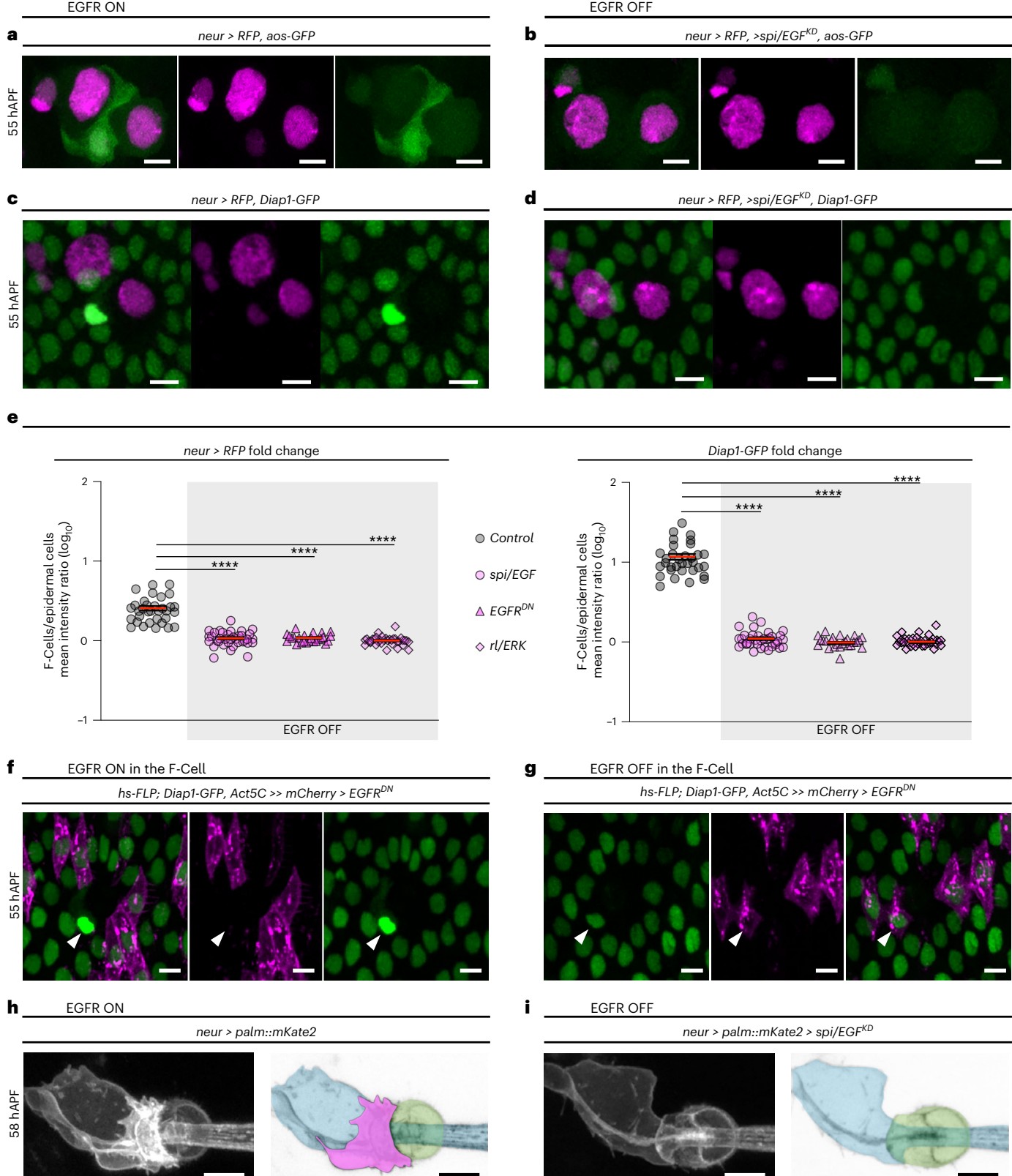

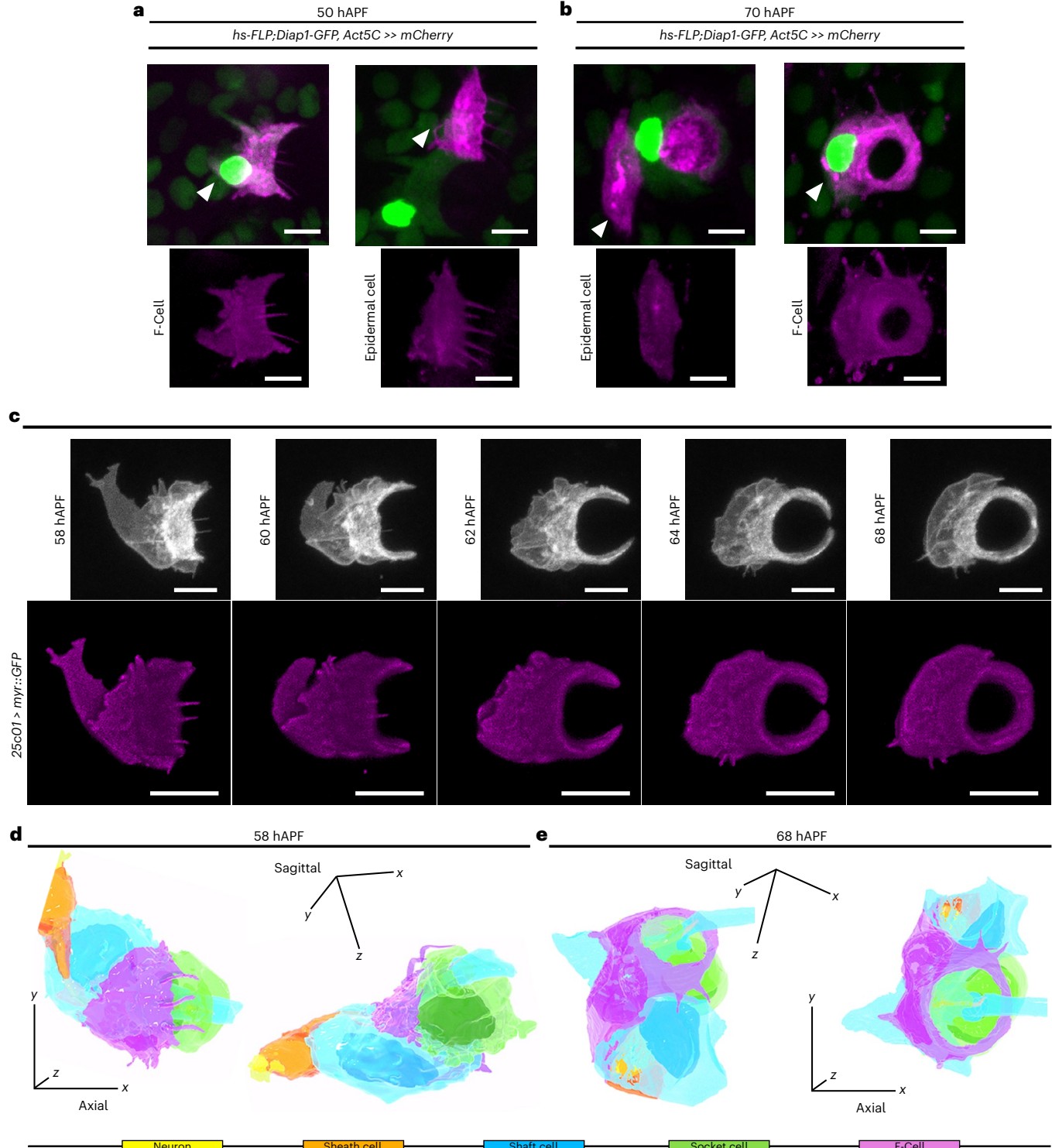

**Fig. 4 | Morphological differentiation of the F-Cell. a**, Time-lapse imaging showing the morphological differentiation of the F-Cell after its specification. Top row: F-Cell morphology labelled by the expression of a membrane-localized GFP under control of *25cO1-GAL4* (*25cO1 > myrGFP*; grey). See also Supplementary Video 3. Bottom row: volume rendering of the F-Cell shape. **b,c**, 3D rendering of the tactile bristle and the F-Cell at 58 hAPF (**b**) and at 68 hAPF (**c**) from SBF-SEM data. The 3D-(*xyz*) coordinate system indicates the angular view of the bristle and F-Cell with respect to the epidermal tissue plane. See Supplementary Videos 4–6 and Supplementary Table 2. Results are representative of three independent experiments. Scale bars, 5 μm. Full genotypes are listed in Supplementary Table 1.

### Epidermal F-Cells ensheath differentiated bristles

We next investigated whether specified F-Cells have similar or distinct morphologies to the epidermal cells in the supporting epidermis. At 50 hAPF, the onset of adult cuticle secretion by the abdominal epidermal cells[34,35], F-Cells and epidermal cells shared similar morphologies (Fig. 4a), including the presence of apical trichomes and an elongated trapezoidal shape[36]. At 70 hAPF, however, while epidermal cells retained their polygonal shape, F-Cells appeared markedly different, adopting a

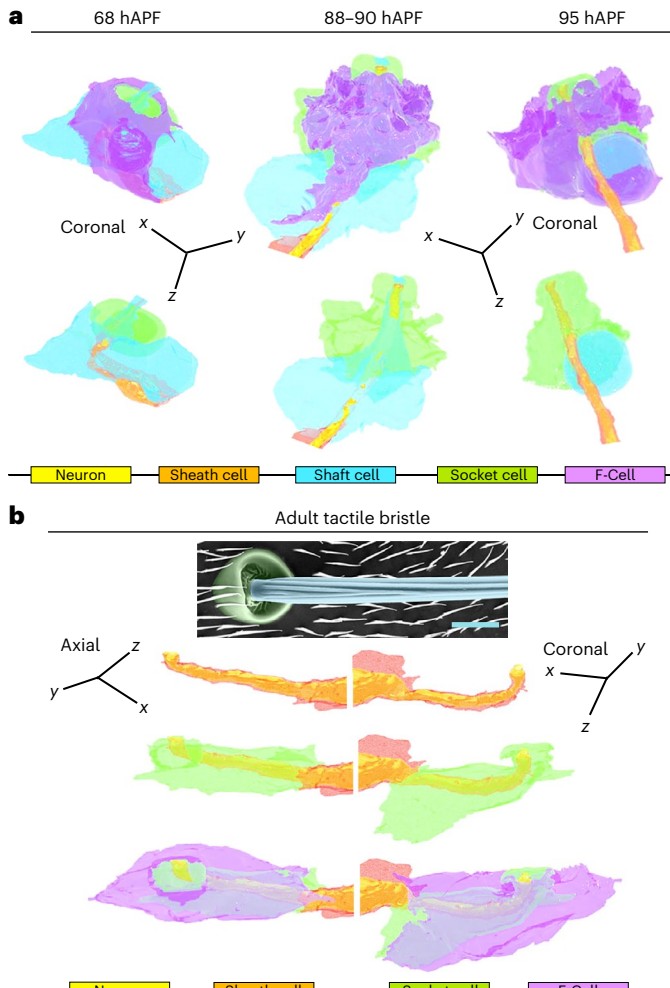

**a**

68 hAPF 88–90 hAPF 95 hAPF

Coronal

| Neuron | Sheath cell | Shaft cell | Socket cell | F-Cell |

**b**

Adult tactile bristle

Axial Coronal

| Neuron | Sheath cell | Socket cell | F-Cell |

**Fig. 5 | Epidermal F-Cells ensheath the tactile bristles. a**, 3D rendering of the tactile bristle between 68 hAPF and 95 hAPF visualized with the F-Cell (top) or showing bristle cells only (bottom). Note that the F-Cell is wrapping around the socket cell, while contacting both the shaft and socket cells more basally. See Supplementary Videos 7 and 8. **b**, Top: SEM image showing the external features of the adult bristle. The cuticular socket and hair shaft are false coloured in green and light blue, respectively. Bottom: 3D rendering of the cellular structures of the adult tactile bristle visualized with the F-Cell, showing exclusively bristle cells, or the mechanosensory neuron and enveloping sheath cell membrane. The F-Cell is wrapping around the socket cell, and the socket cell is wrapping around the sheath cell encapsulating the neuron dendrite. See also Supplementary Videos 9 and 10 and Supplementary Table 2. Results are representative of two (88–90 hAPF and 95 hAPF) or three independent experiments. Scale bar, 5 μm. Full genotypes are listed in Supplementary Table 1.

ring-like shape encircling the socket cell (Fig. 4b). These observations suggest that F-Cells undergo a morphological differentiation during adult cuticle deposition. To study F-Cell morphology in more detail, we screened the expression patterns in the PNS of enhancer-driven GAL4 lines from the *Drosophila* FlyLight collection[37] for reporter lines that were specifically expressed in F-Cells. We identified the *25c01-GAL4* line, derived from the *aos* locus, as an unambiguous marker of F-Cells (Fig. 4c and Extended Data Fig. 7a). We next monitored the shape of the F-Cell by time-lapse imaging, revealing the dramatic morphological changes experienced by this cell after its specification (Fig. 4c, Extended Data Fig. 7b and Supplementary Video 3). Between 58 and 62 hAPF, F-Cells progressively acquired a crescent-like shape next to the socket cells, while extending dynamic protrusions sub-epidermally

(Fig. 4c). By 64 hAPF, the F-Cell extended its contact with the socket cell further, encircling it entirely by 68 hAPF (Fig. 4c) and moving more basally as the adult cuticle was formed (Extended Data Fig. 7c). These data indicate that differentiating F-Cells become intimately associated with the bristle by progressively wrapping around the tactile organ.

To characterize the morphology of the F-Cell and its interaction with the tactile organ at higher resolution, we performed 3D rendering of the bristle structure using serial block face scanning electron microscopy (SBF-SEM)[38] at different timepoints during differentiation (Extended Data Fig. 8a, Supplementary Table 2 and Methods). 3D rendering at 58 hAPF shows that the F-Cell is in contact with the anterior portion of the differentiating socket cell and, more basally, with the outgrowing hair shaft and the shaft cell body (Fig. 4d, Extended Data Fig. 8b,c and Supplementary Videos 4 and 5). At 68 hAPF, the F-Cell has extended its contacts with the socket cell, encircling part of its cytoplasm (Fig. 4e, Extended Data Fig. 8d and Supplementary Video 6). A characteristic feature of the non-neuronal cells in developing insect bristles is their concentric organization around the sensory neuron dendrite[6,10]. By 68 hAPF, the F-Cell shares this feature, being the outermost cell that concentrically surrounds the bristle (Fig. 5a and Extended Data Fig. 8d). At 88–90 hAPF, the F-Cell and the socket cell have expanded their mutual contacts, while the shaft cell is retracting away from the hair shaft base (Fig. 5a, Extended Data Fig. 9a and Supplementary Video 7). By the end of pupal development, the shaft cell has undergone cell death, and its cytoplasm has fully retracted from the hair shaft base and the dendrite, which remains concentrically surrounded by the sheath cell, the socket cell and the F-Cell (Fig. 5a, Extended Data Fig. 9b,c and Supplementary Video 8).

To determine the cellular structure of the mature bristle and association with the F-Cell, we performed 3D rendering of the adult bristle (Fig. 5b, Supplementary Videos 9 and 10 and Supplementary Table 2). Remarkably, the F-Cell association with the bristle persists in the adult mechanosensory organ, where the neuron dendrite and sheath cell membrane are concentrically surrounded by the socket and the F-Cell membrane (Fig. 5b and Extended Data Fig. 9d). Together, these data demonstrate that the F-Cell associates with the bristle from late differentiation through adulthood.

## Epidermal F-Cells are required for touch sensing

Adult bristles mediate robust mechanotransduction upon direct touch. Touch-evoked deflections of the hair shaft towards the epidermal surface are transmitted via the dendrite sheath to the dendritic membrane, eliciting action potential trains through the depolarization of the neuron[7,10,39]. Since the F-Cell associates with the adult tactile bristle, we asked whether F-Cells were required for mechanotransduction. To address this, we first performed electrophysiological recordings from control bristles and analysed their responses upon touch (Extended Data Fig. 10a,b). At rest, the tactile bristles display a positive transepithelial potential (TEP) (Fig. 6a), due to the ionic gradient established within the bristle relative to the surrounding epithelium[10,39–41]. The neuronal response upon hair shaft deflection is characterized by a sharp change in the TEP towards negative values, quantified as the mechanoreceptor potential (MRP)[8] of the bristle (Fig. 6a and Extended Data Fig. 10a,b). We next performed recordings from bristles in which F-Cell specification was genetically prevented (*neur > spi/EGF^{KD}* versus *neur > GFP^{KD}* controls, Fig. 6b). In this condition, both the structure of the tactile bristle and its innervation pattern appeared unaltered by the absence of the F-Cell (Extended Data Fig. 10c,d), allowing the measurement of the resting TEP and touch-evoked MRP (Fig. 6b). We found that the mean resting TEP was decreased, and the MRP was severely reduced in tactile bristles lacking the F-Cells relative to controls (Fig. 6b and Extended Data Fig. 10e), revealing compromised neuronal depolarization upon hair deflection. To further test whether this was a consequence of the lack of the F-Cell, we performed recordings from tactile bristles in which the F-Cell was genetically ablated

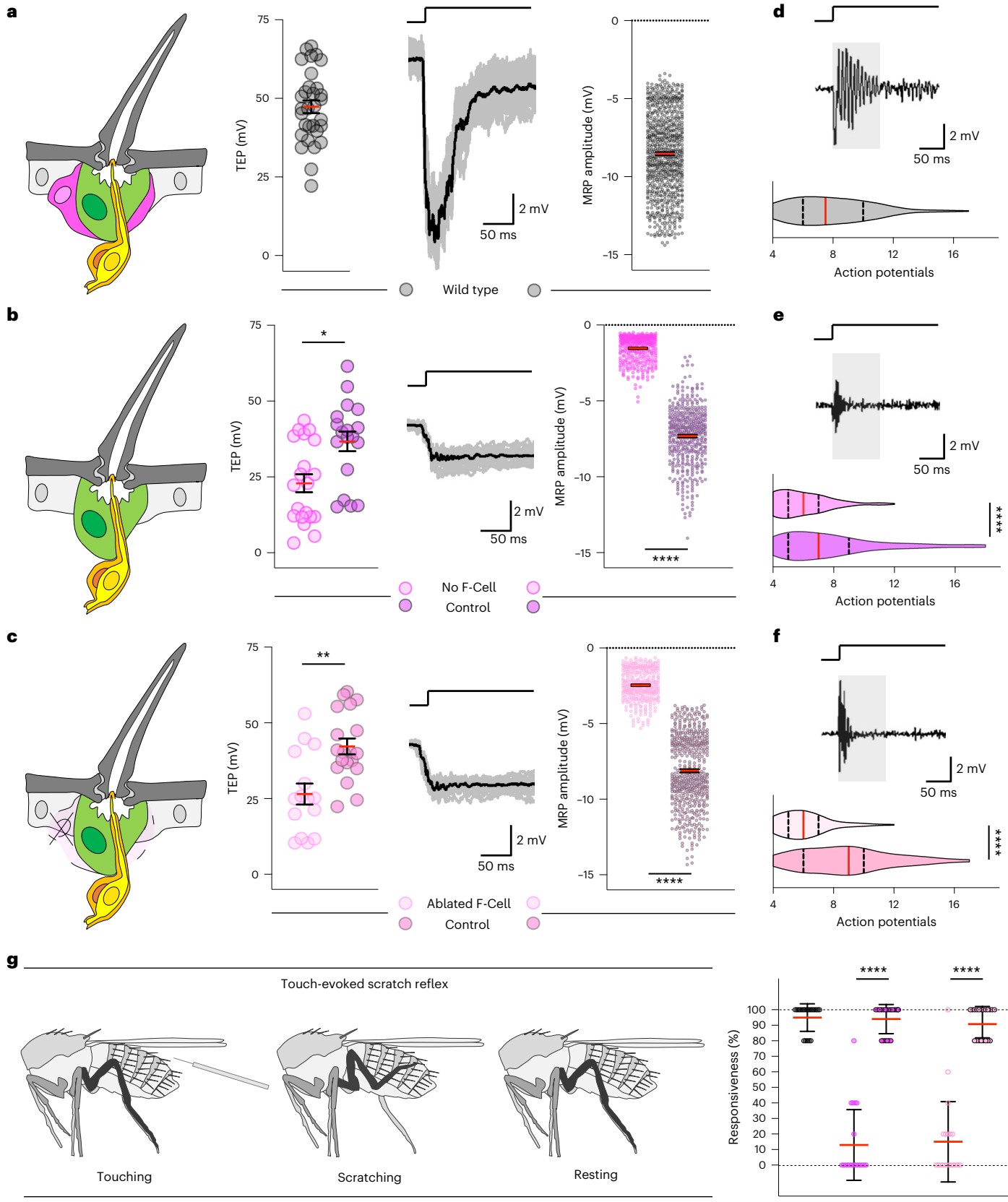

after its specification by overexpressing the pro-apoptotic gene *grim* within the F-Cell (*25c01 > grim* versus *25c01 > GFP* controls, Fig. 6c). Also in this case, touch-evoked MRP was markedly reduced relative to controls (Fig. 6c and Extended Data Fig. 10f). Therefore, tactile bristles that lack F-Cells show abnormal mechanotransduction.

We next tested if F-Cells are required for neuronal firing upon touch and quantified the action potentials fired at the onset of the tactile stimulus as a direct measure of sensory neuronal firing. Unlike control bristles, in which mechanosensory neurons show robust firing at stimulus onset (Fig. 6d and Extended Data Fig. 10b), bristles lacking the

**Fig. 6 | Tactile bristles require epidermal F-Cells for touch sensing. a**, Left to right: diagram of the adult tactile bristle and the F-Cell, quantifications of the TEP at rest, voltage traces with the averaged response of the neuron upon hair shaft deflection in black, and quantifications of the MRP peak amplitudes from stimulated wild-type *Ore-R* bristles (*n* = 31 TEPs and 762 MRPs pooled from 15 flies and three independent experiments). **b,c**, Left to right: diagram showing the adult tactile bristle without the F-Cell, quantifications of the TEP at rest, voltage traces with the averaged response of the neuron in black, and quantifications of MRP peak amplitudes from stimulated bristles lacking the F-Cell (**b**: *n* = 20 TEPs and 380 MRPs versus *n* = 18 TEPs and 460 MRPs paired controls pooled from eight flies and three independent experiments for each condition) or from bristles with ablated F-Cells (**c**: *n* = 15 TEPs and 349 MRPs versus *n* = 19 TEPs and 538 MRPs paired controls pooled from ten flies and three independent experiments for each condition). **d**, Action potential (AP) burst firing and AP count at stimulus onset (grey box) in wild-type *Ore-R* (*n* = 13 bristle and 1,016 APs pooled from nine flies and three independent experiments for each condition). **e,f**, AP burst firing and

AP count in bristles lacking the F-Cell (**e**: *n* = 13 bristles and 745 APs versus *n* = 11 bristles and 855 APs paired controls pooled from 10 flies and three independent experiments for each conditions), or in bristles after ablation of the F-Cell (*n* = 11 bristles and 658 APs versus *n* = 9 bristles and 802 APs paired controls pooled from 11 flies and three independent experiments for each condition). **g**, Diagram of the touch-evoked scratch reflex assay and quantifications of the responsiveness upon touch in controls versus flies lacking the F-Cell (*n* = 5 stimuli per fly, 20 flies in two independent experiments per condition). Data are mean (red bar) ± s.e.m. (black bars) (**a–c**) or mean ± s.d. (black bars) (**g**). In violin plots, the kernel density distribution of the data is shaped around the central median, extending to the 25% and 75% quartiles (dashed lines) up to the maximum and minimum values. The two-tailed unpaired Kolmogorov–Smirnov test; \**P* < 0.05, \*\**P* < 0.01 (TEP **a–c**) or Mann–Whitney test; \*\*\*\**P* < 0.0001 (**d–g**), and one-way ANOVA with Kruskal–Wallis test; \*\*\*\**P* < 0.0001 (MRP **a–c**) were performed. Results are representative of three independent experiments. Full genotypes are listed in Supplementary Table 1. Numerical data and exact *P* values are available in source data.

F-Cell displayed decreased neuronal excitability upon hair deflection (Fig. 6e,f). Therefore, although the bristle neuron can still discharge action potentials in the absence of the F-Cell, its sensitivity to hair deflection is markedly reduced. Taken together, these data indicate that F-Cells are necessary to amplify touch-evoked stimuli so that they result in robust bristle neuron firing. We next sought to determine whether F-Cell association with the bristle impacts somatosensory behaviour upon touch-evoked stimuli. When the hair shaft of a tactile bristle is stimulated by direct touch, a scratch reflex is elicited by the thoracic circuit of the ventral nerve cord, even in the absence of descending inputs from the brain[39,42,43]. Stimulation of the abdominal tactile bristles invariably elicits third leg movements towards the stimulated hair shaft[39]. We therefore tested the ability of flies with and without F-Cells to elicit a scratch reflex upon touch of the tactile bristles (Fig. 6g). Notably, the response was robust in all controls, but severely reduced when tactile bristles lacking the F-Cell were stimulated (Fig. 6g). Thus, F-Cells are critical players in mediating touch sensing in *Drosophila*.

## Discussion

Here, by examining the cellular assembly of the mechanosensory bristles that cover much of the fly body surface, we identify the epidermal F-Cell as a previously undescribed cell type that associates with tactile bristles and influences their neurophysiological signature. Our findings demonstrate that the cells within the tactile bristle, which are all related by lineage[11], are not sufficient to assemble a fully functional touch-sensitive organ in the adult fly. We show that a signalling dialogue with the resident epidermis is essential for selecting F-Cells and for the acquisition of their specialized morphology.

Our work establishes that F-Cells are specialized epidermal cells that ensheath the mature tactile bristle and are required for normal neuronal sensitivity upon hair deflection. How is sensitivity to touch controlled by the F-Cell? The altered electrophysiological signature of bristles lacking the F-Cell indicates that the mechanism could be mechanical, electrochemical or both. A possible scenario is that the F-Cell, by ensheathing the tactile bristle, might apply strain to the neuronal tip upon hair deflection, which in turn would facilitate the opening of mechanically gated ion channels around the sensory ending[7,44]. Therefore, F-Cells would be physically coupling touch-evoked hair deflections to neuronal depolarization, as has been proposed for the circumferential and lanceolate endings that complex with the mouse hair follicles[2]. Alternatively, F-Cells might modulate the ionic milieu that surrounds the sensory ending, acting as a glial-like cell, as has been proposed for other non-neuronal cells associated with PNS neurons[45].

Though the molecular details underlying F-Cell interplay with the tactile bristles remain to be determined, our findings raise the exciting possibility that F-Cells may be analogous to specialized non-neuronal cells associated with cutaneous mechanosensory organs in

other animal species. For example, the epidermal Merkel cells that cluster around the guard hair follicle in mice display a unique crescent-like shape that distinguishes them from surrounding keratinocytes[46,47]. Thus, the acquisition of a unique morphology, such as we observed for the F-Cell, is clearly an important feature that probably allows non-neuronal cells to fulfil their key roles in mechanotransduction.

In summary, our findings support a model in which the physiological properties of PNS neurons are strongly influenced by a unique combination of specialized non-neuronal cells in their surroundings[1,3]. We propose that the tactile bristle/F-Cell association uncovered in *Drosophila* can serve as a powerful model to study how non-neuronal cells shape tactile perception in vivo.

## Online content

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

## Methods

### *Drosophila* husbandry

Fly stocks were reared on a standard yeast-cornmeal-agar medium (360 g agar, 3,600 g maize, 3,600 g malt, 1,200 ml molasses, 440 g soya, 732 g yeast extract, 50 l water and 280 ml of acid mix−500 ml propionic and 32 ml of orthophosphoric acid). All experimental flies were kept in incubators under stable humidity (60%) and on a controlled 12 h light/dark cycle. Flies were kept at 18 °C, 25 °C or 29 °C depending on the specific experiment. Pupae were staged according to Bainbridge and Bownes[48] and timed employing puparium formation as a reference (hAPF).

### *Drosophila* strains

The GAL4/UAS and LexA/Aop binary systems[49,50] were used to restrict or modulate gene expression. The following drivers were used: *neur^p72-GAL4* (ref. [51]), referred to as *neur-GAL4*), *GMR25c01-GAL4* (BDSC: 49115, referred to as *25c01-GAL4*), *57c10-GAl4* (BDSC: 39171, referred to as *nSyb-GAL4*), *57c10-LexA* (BDSC 52817, referred to as *nSyb-LexA*), *Act5C-FRT > y > FRT-GAL4* (BDSC: 3953), *Tub-GAL80, FRT40A* (BDSC 5192), and *hsp70-FLP122* (BDSC 23647, referred to as *hs-FLP*). The following reporters were used: *UAS-mCD8::GFP* (BDSC: 5130), *UAS-mCD8::Cherry* (BDSC: 27391), *UAS-H2B::RFP*[52], *UAS-RFP.nls* (BDSC: 31417), *UAS-GFP.nls* (BDSC: 4776), 10X*UAS-myr::GFP* (BDSC: 32197), *UAS-GFP::CLIP-170* (ref. [53]), *UAS-pon::GFP*[54], *UAS-palm-mKate2* (BDSC: 86540), *13XLexAop2-IVS-myr::RFP* (gift from I. Salecker), *UAS-H* (gift from S. Bray), *UAS-H-RNAi* (TRiP: 34703, referred to as *H^KD*), *UAS-spi-RNAi* (TRiP: 34645, referred to as *spi^KD*), *UAS-EGFR-DN* (BDSC 5364), *UAS-GFP-VALIUM10/20* (TRiP: 35786, referred to as *GFP^KD*), *UAS-grim* (gift from A. Gould), *aos^2-GFP* (ref. [30]) (referred to as *aos-GFP*), *Diap1^4.3-GFP*[21], *Ubi-GFP.nls* (BDSC: 5629), *EGFR::GFP*[55], *Jupiter::GFP* (BDSC: 6825), *E(Spl)ma-GFP*[14], *Su(H)ASE5-GFP* (BDSC: 58449), *Tub-GAL80^ts* (BDSC: 7108) *E-Cad::Td-tomato*[56] and *E-Cad::mKate2* (gift from Y. Bellaiche). The following mutant alleles were used: *rl^l* (BDSC: 386) and *rl^10a* (BDSC: 742), and *spi^SC1 FRT40A* (gift from V. Fernandes). *Oregon R* (*Ore-R*) was used as wild-type strain. Multiple drivers and reporters were stably combined by standard recombination methods. For a list of the full genotypes used in this study, see Supplementary Table 1.

### Temporal control of gene expression

Temporal control of gene expression was used to label the shape of individual cells within the abdominal epidermis with the FLP-Out-GAL4 method[57]. The FRT-flanked stop cassette of the transgene *Act5C-FRT > y > FRT-GAL4, UAS-mCD8::Cherry* was excised (referred to as *Act5C»mCherry*) by activating FLP expression at 16 hAPF using a 3 min heat shock at 37 °C in a water bath. Randomly labelled F-Cells were identified by elevated expression of the *Diap1-GFP* reporter and stereotypical position relative to the socket cell. F-Cell morphology was compared with the morphology of randomly labelled epidermal cells co-expressing membrane-localized *mCD8::Cherry* and basal levels of *Diap1-GFP*. The same approach was used to induce inactivation of EGFR signalling in a *Diap1-GFP* background via *UAS-EGFR^DN* expression. F-Cell specification was scored by levels of expression of *Diap1-GFP* within *mCD8::Cherry* expressing cells (that is, EGFR ON) and unlabelled cells (that is, EGFR OFF) at 55 hAPF. Shaft-to-socket or socket-to-shaft cell transformations in the *Diap1-GFP* background were induced via *H^KD* or *UAS-H* expression and the FRT-flanked stop cassette was excised at 24 hAPF using a 5–7 min heat shock at 37 °C using a water bath. Temporal control of gene expression was also used to silence *spi* expression in the shaft cells with the MARCM method[58] in a *neur > RFP, Diap1-GFP* background. Mitotic recombination between FRT sites in flies of the genotype *hsFLP; Tub-GAL80, FRT40A/spi^SC1 FRT40A; neur > RFP, Diap1-GFP* was induced by activating FLP expression at 20 hAPF using a 40 min heat shock at 37 °C in a water bath. Expression of *neur > RFP* was used to scored cells with silenced *spi* activity. Temporal control of gene

expression was also used to genetically ablate the F-Cells. The ubiquitously expressed temperature sensitive *GAL80* transgene (*Tub-GAL80^ts*) was used in combination with the *UAS-grim* (or *UAS-GFP* for controls), the *25c01-GAL4* transgenes, and for the *UAS-EGFR^DN* expression with *neur-GAL4*. Flies were crossed and reared at 18 °C (permissive temperature) until 40 hAPF and transferred at 29 °C (restrictive temperature) until adulthood. RNA interference experiments for gene knockdowns under control of *neur-GAL4* were also performed at 29 °C to increase the strength of transgene expression.

### Live Imaging with confocal microscopy

Staged pupae of the desired genotypes were dissected and mounted as previously described[59]. Briefly, the pupae were dissected from the puparium using forceps. Naked pupae were transferred to a glass-bottom dish containing small drops of gas-permeable halocarbon oil to improve optics during imaging with oil immersion objectives. Pupae were imaged on their dorsal side using an inverted microscope Zeiss LSM 880, Plan-Neofluar 40×/1.3 (numerical aperture) oil immersion objective and ZEN software (version 2.1). Solid state and argon lasers were used for detection of fluorescence signals in confocal mode. Bright-field images were collected using the transmission light T-PMT detector. For time-lapse recordings, the time interval was set at 5 min. Laser intensity was kept to a minimum to prevent photobleaching. Pupae were cultured after imaging showing normal development.

### Single cell ablation with two-photon microscopy

High-power femtosecond lasers can induce damage to cells and tissues by a process called optical breakdown[60] that ultimately leads to thermal and mechanical disruption of the cell or tissue to which they are directed. We used this principle to ablate epidermal or tactile bristle cells at the desired time during pupal development by tuning the two-photon laser to 780 nm wavelength under software control (ZEN 2.1, Zeiss). A circular region of 36 × 36 pixels was digitally made to fit the centre of the cell nucleus and ablated with a dwell time of 1.58 µs per pixel at 70% laser power within a stack of 16 slices, for one iteration, under software control. This spatial and temporal focal confinement of the femtosecond laser ensured that the thermal damage was restricted to the cell of interest. Image acquisition before, and after, cell ablation was at 10 s intervals with a Plan-Neofluor 40×/1.3 (numerical aperture) oil immersion objective from an inverted microscope with laser scanning unit LSM 780 (Zeiss), coupled with a two-photon femtosecond laser (Chameleon-XR Ti-Sapphire; Coherent Inc.). Damage to the targeted cell was observed immediately after the operation (that is, 10 s after ablation). The subsequent cell death and behaviour of remaining cells were monitored by live imaging in confocal mode with a time interval of 5 min.

### Image processing and quantification of fluorescence intensity

All images and time lapses were visualized and processed using ImageJ/Fiji (https://fiji.sc)[61]. All acquired stacks were displayed in 2D using the Maximum Intensity option, except for bright-field images, where the Minimum Intensity option was applied. Some confocal stacks were displayed in 3D using the 3D-Viewer plugin or the 3D projection tool. Quantification of GFP and RFP levels of expression were performed using the Analyse tool in Fiji. Areas of identical size overlaying epidermal F-Cells or surrounding epidermal cells were selected manually across experiments/genotypes. After background subtraction, mean intensity for each area was measured per experiment/genotype.

### SBF-SEM

Serial block face scanning electron microscopy (SBF-SEM)[38,62] was used to reconstruct the morphologies of the tactile bristle and the F-Cell 3D.

**Sample preparation for SBF-SEM.** Wild-type *Ore-R* pupae were collected, peeled off the outer pupal case[59] and transferred in freshly

prepared fixation buffer (4% paraformaldehyde and 2.5% glutaraldehyde in 0.1 M phosphate buffer at pH 7.4). The abdomens from naked pupae or adult flies were dissected in the pre-fixation buffer using surgical scissors and internal organs were manually removed using forceps and incubated in the pre-fixation buffer for 1 h at room temperature (RT). The abdominal epidermis from pupal samples were post-fixed and stained using a modified version of the National Center for Microscopy and Imaging Research method[63], where samples were incubated in reduced osmium (1% osmium/1.5% potassium ferricyanide at 4 °C) for 1 h and stained with 1% thiocarbohydrazide for 20 min at RT, followed by 2% osmium tetroxide for 30 minutes at RT and an overnight incubation in 1% uranyl acetate. Samples were then en bloc stained with lead aspartate (pH 5.5) for 30 min at 60 °C, washed in distilled water (3 × 5 min) and dehydrated using a graded ethanol series (20%, 50%, 75%, 90%, 100% × 2, 20 min each). Samples were infiltrated in Durcupan (44610-1EA, Sigma-Aldrich) 1:1 resin:ethanol overnight, 100% resin for 24 h and polymerization at 60 °C for 48 h. Excess resin was removed[64] and samples were mounted on three-view pins (10-006002-50, Labtech) with the hair shafts facing up and polymerized at 60 °C for 48 h. For the 95 hAPF and adult samples, a progressive lowering of temperature and low-temperature staining protocol was used to enhance the contrast in samples with thicker cuticles[65]. Sample were post-fixed in 1% osmium tetroxide at RT for 40 min, incubated in 1.5% potassium ferrocyanide for 2 h at 4 °C, and washed. This was in distilled water (3 × 10 min). Samples were then transferred to 1% thiocarbohydrazide for 15 min at RT followed by en bloc staining in lead aspartate for 30 min at 55 °C and 1 h at RT. Samples were transferred to an automatic freeze substitution chamber (AFS2, Leica) and dehydrated in a graded series of acetone while temperature was progressively lowered from 0 °C to −25 °C (10% from 0 °C to −5 °C, 30% from −5 °C to −10 °C, 50% from −10 °C to −15 °C, 70% from −15% to −20 °C, 90% from −20 °C to −25 °C and 97% at −25 °C, 20 min each step). Samples were subjected to low temperature en bloc staining of 1% osmium, 0.2% uranyl acetate in 97% acetone for 30 h at −25 °C followed by the warming to RT over 5 h. Before imaging, the embedded samples, were subjected to microscopic X-ray computed tomography on a Versa 510 (Zeiss) using a 4× objective and an operating voltage of 40 kV and binning 5 for proper orientation of the block. Following microscopic X-ray computed tomography, the embedded samples were trimmed using a glass knife on an ultramicrotome (UC7, Leica) to the correct orientation, removed from the top of the block using a razor blade[66] and mounted onto three-view pins using silver epoxy (604057, CW2400 adhesive, Farnell) that was then polymerized at 60 °C for 1 h. Samples were then trimmed down further using a glass knife and ultramicrotome to an area approximately 800 µm × 800 µm in size.

**SBF-SEM imaging.** Samples were sputter coated with a 10 nm layer of platinum (Q150S, Quorum Technologies) and loaded into a 3View2XP (Gatan) attached to a Sigma VP SEM (Zeiss) with focal charge compensation (FCC, Zeiss), and data were collected using a BSE detector (3View detector, Gatan). Imaging parameters for each stage of tactile bristle differentiation can be found in Supplementary Table 2. The morphology of the neuronal dendrite from bristles lacking the F-Cell was visualized in the genotype *neur > spi/EGF^{KD}*, and no obvious defects were detected in the neuron and surrounding sheath cell membrane relative to wild-type controls.

**3D shape reconstruction.** Each dataset was aligned and the cells of interest were manually segmented using the area list function of the TrakEM2 Plugin in Fiji[67]. The area lists of the 3D volumes were exported as wavefront files for surface rendering using the 3D viewer plugin of Fiji[68]. The wavefront files were imported into the open-source 3D modelling software Blender (Blender 2.9, www.blender.org) and scaled in *x*, *y* and *z*, keeping the proportions consistent with the imaging conditions (for example, if *x* and *y* had a resolution of 10 nm and *z* was 50 nm, the

model would be scale *x* 0.001, *y* 0.001 and *z* 0.005). Texture was then applied to each model to give the desired colour and transparency. For each stage of development, three types of animation were generated using the timeline option of Blender: animations of the bristle around one axis ('rotation' videos) and two distinct animations with the progressive appearance/disappearance of the cells of the bristle ('vanishing' videos and 'peeling' videos).

## SEM

*Ore-R* adult female flies within 1 week after eclosion were used for visualizing the external morphology of the bristle using SEM. Anesthetized flies of the desired genotypes were deprived of their wings and legs, washed in standard saline solution three times, and stored in absolute ethanol (99.8%, Sigma-Aldrich) at RT. Dehydrated samples were critical point dried (EM CPD300, Leica Microsystems) and mounted with their dorsal side up on a stub using carbon tape and silver paint. Samples were sputter coated (Q150RS Plus, Quorum Technologies) with platinum at approximately 3 nm thickness. Images were collected on a JCM-6000Plus (JEOL) SEM using the Everhart-Thornley secondary electron detector at 10 kV.

## Electrophysiology

Extracellular recording was used to gain electrical access to the tactile bristle at rest and upon mechanical displacement of the hair shaft as previously described[39] with some modifications. Young adult female flies (3–5 days after eclosion) were deprived of their wings and legs and kept immobilized using insect pins inserted into the body extremities. The hair shaft from individual abdominal bristles was clipped to approximately 30% of its full-length using microdissection scissors. For all recordings, a pair of silver/silver chloride (Ag/AgCl) wires inserted into patch pipettes made from borosilicate glass with a P-2000 puller (Sutter Instrument) were used as reference and recording electrodes (1–2 MΩ resistance). The reference electrode was filled with the reference saline solution (2 mM K$^+$, 128 mM NaCl, 0.5 mM Ca$^{2+}$, 4 mM Mg$^{2+}$, 35 mM glucose and 5 mM HEPES, pH 7.1) and inserted into the haemolymph space bathing the epithelium. The recording electrode was filled with the recording saline solution (121 mM K$^+$, 9 mM NaCl, 0.5 mM Ca2 +, 4 mM Mg$^{2+}$, 35 mM glucose and 5 mM HEPES, pH 7.1) and slipped over the cut end of individual bristles, thus gaining access to the mechanoreceptor lymph and making a circuit across the sensory epithelium through the hollow hair shaft[10,39,41]. The voltage offset was corrected by placing both electrodes into the saline bath before recording from the cut end of the bristle shaft. The TEP, or voltage difference between the bristle and the supporting epithelium, was measured as the voltage difference between the two electrodes at rest. The MRP amplitude was subsequently recorded as a change in the TEP upon mechanical displacement of the bristle shaft. All MRPs were evoked by 30 µm deflections of the recording electrode towards the body (that is, the preferential direction[7,69]) by software-controlled mechanical movement of a PatchStar micromanipulator (Scientifica). All recordings were made in current-clamp mode using a MultiClamp-700B amplifier (Molecular Devices). Data were low pass filtered at 50/60 Hz (hum silencer), sampled at 2 kHz and digitized with the Axon Digidata 1550A A/D board (Molecular Devices). Recordings were stored as axon binary files and analysed offline using Clampfit software (v10.7, Molecular Devices). Traces were low pass filtered using a Butterworth eight-pole filter with cut-off frequency of 250 Hz. MRPs were automatically extracted using a template search method, and MRP peak amplitudes were subsequently computed from each trace (multiple MRPs for each bristle were recorded from each trace). To quantify action potential firing in response to bristle deflection, MRP traces were first bandpass filtered (50 Hz-RC single pole high-pass, Butterworth eight-pole low-pass with cut-off frequency of 250 Hz) to enable spike detection. The number of action potentials at stimulus onset (that is, higher firing rate of the bristle) were counted using the Clampfit burst

detection algorithm (minimum cut-off of four events, intra-event interval within 120 ms). Recordings from multiple bristles from a minimum of eight adult female flies for each genotype were used in this study.

## Behaviour

Touch-evoked scratch reflex assay[39,42] was performed to evaluate motor reactivity upon mechanical stimuli of the abdominal bristles in controls versus experimental flies. Experiments were performed as previously described[70] with some variations. Briefly, young female flies (3–5 days post eclosion) were anesthetized on ice and their head was removed using microdissection scissors. Headless flies were allowed to recover their normal stance before the assay. An insect pin was then attached to the thorax to hold each fly in a fixed position during the test. Touch-evoked sweeping of the third leg in response to gentle touch of the abdominal bristles was assayed using a glass pipette equipped with a curved 0.1 mm pin on its tip. Each fly was stimulated five times every 5–8 s (100 stimuli were counted for each genotype). The behavioural responses were recorded in continuous mode with a high-speed camera (Zeiss Stereo Microscope), and the number of scratch responses was counted offline.

## Statistics and reproducibility

Sample sizes were chosen according to common standards. All results were obtained from at least three independent experiments unless otherwise stated in the figure legends. Data collection and analysis were neither performed blind to the conditions of the experiments nor randomized. Statistics were performed with the PAST 4.04 software[71]. Statistical tests were chosen on the basis of data distribution normality, which was tested using Shapiro–Wilk test. No statistical methods were used to pre-determine the sample size. Either unpaired two-tailed parametric Student's $t$-test, two-tailed non-parametric Kolmogorov–Smirnov and Mann–Whitney tests, or one-way analysis of variance (ANOVA) with the non-parametric Kruskal–Wallis test were performed for comparing two groups of data, as indicated in the figure legends. Differences were considered significant when $P < 0.05$ and indicated as follows: $*P < 0.05$; $**P < 0.01$; $***P < 0.001$; $****P < 0.0001$; $P > 0.05$ not significant (NS). Exact $P$ values can be found in source data. No data points were excluded from the analyses. Data were plotted using GraphPad Prism 9 (GraphPad Software). Dot plots show the mean, and error bars represent the standard error of the mean (s.e.m.) or the standard deviation (s.d.), as indicated in the figure legends. Violin plots show the kernel density distribution of the data around the central median, extending to the 25% and 75% quartiles (dashed lines) up to the maximum and minimum values. In all figures, $n$ denotes the number of bristles, cells, clones or other indicated parameters that were analysed for each genotype. Figure panels and diagrams were created using Photoshop and Illustrator (Adobe).

## Reporting summary

Further information on research design is available in the Nature Portfolio Reporting Summary linked to this article.

## Data availability

The data supporting the findings of this study are available within this paper, extended data, source data and supplementary information. Additional information is available from the corresponding authors on reasonable request. Source data are provided with this paper.

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

## Acknowledgements

We thank the Bloomington *Drosophila* Stock Center (BDSC), S. Bray, J. Jiang, Y. Bellaiche, B. Stramer, I. Salecker, A. Gould, V. Fernandes and N. Yakobi for fly stocks and FlyBase for *Drosophila* gene annotations. We thank the Crick Advanced Light Microscopy and Electron Microscopy Science Technology Platforms, and the Crick Fly Facility for support. We are grateful to J. P. Vincent, A. Gould, D. Ish-Horowicz, A. Mineo, M. Holder, A. Audibert, C. Desplan and A. Schaefer for comments on the manuscript. We also thank F. Simon for sharing some preliminary observations. This work was supported by a Wellcome Trust Investigator award (107885/Z/15/Z to N.T.) and the Francis Crick Institute, which receives its core funding from Cancer Research UK (FC001317, FC001175), the UK Medical Research Council (FC001317, FC001175) and the Wellcome Trust (FC001317, FC001175). M.G. was funded by Centre National de la Recherche Scientifique and Sorbonne University. I.D. and J.T. were supported by a Welcome Trust Investigator award (209412) to I.D.

## Author contributions

F.M. and N.T. conceived the study. F.M. designed and performed the experiments. F.M. and J.T. performed the extracellular recordings and analysed data with the assistance of I.D. F.M and C.M conducted the EM experiments and analysed the data with the assistance of L.C. M.G. shared unpublished data and provided methodological insights. F.M. and N.T. wrote the manuscript with input from all authors.

## Competing interests

The authors declare no competing interests.

## Additional information

**Extended data** is available for this paper at https://doi.org/10.1038/s41556-023-01110-2.

**Correspondence and requests for materials** should be addressed to Federica Mangione or Nicolas Tapon.

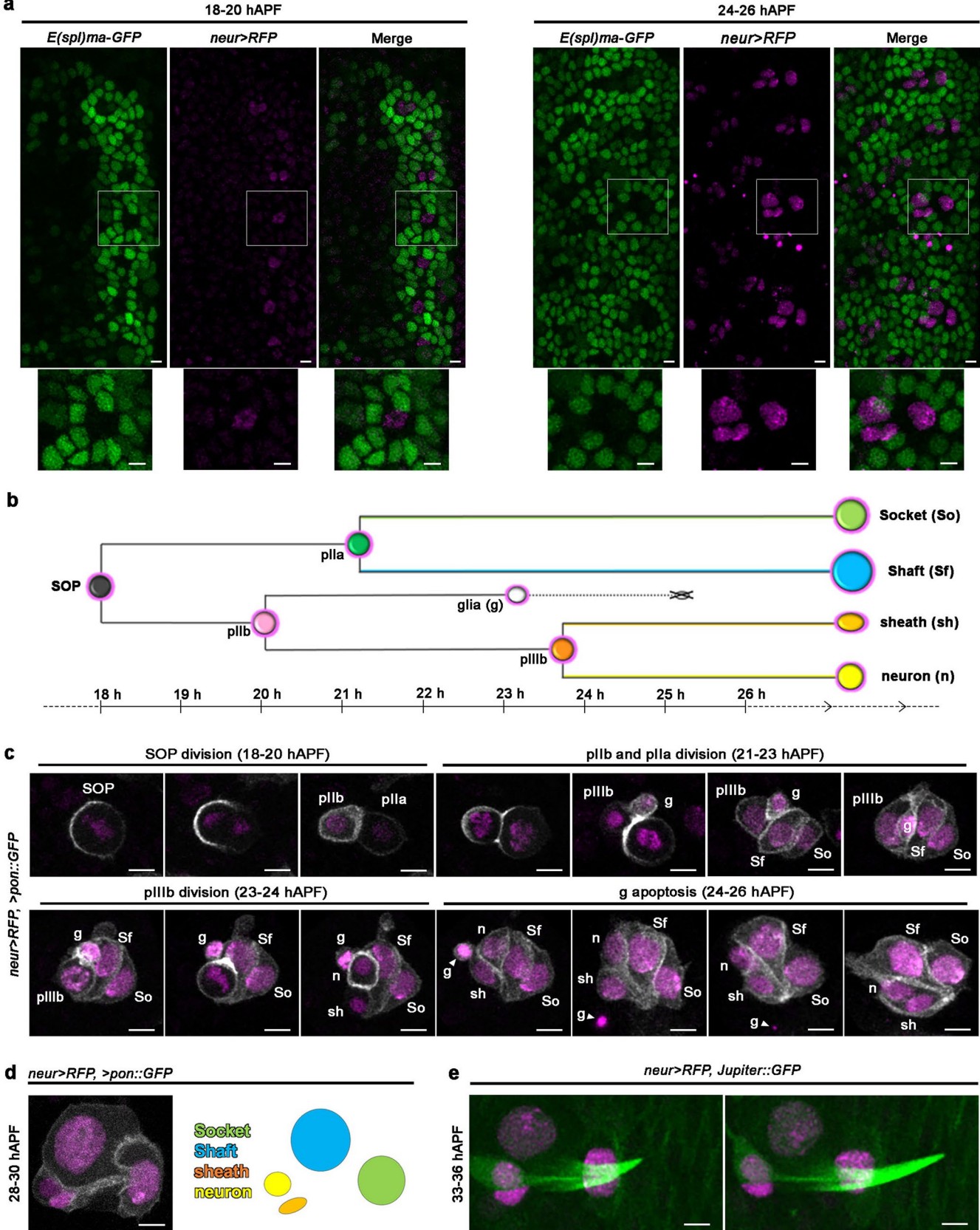

**Extended Data Fig. 1 | See next page for caption.**

**Extended Data Fig. 1 | Early development of the abdominal tactile bristles.**
(**a**) Specification of the tactile bristles within the abdominal epithelium during the first day of pupal development. Left: N-dependent pattern of *E(spl)ma-GFP* (green) and *neur > RFP* (magenta) at the onset of bristle specification. *E(spl)-mαa-GFP* is expressed in the epidermal cells surrounding the SOPs, which are marked by *neur > RFP* expression. Right: the pattern of *E(spl)ma-GFP* and of *neur > RFP* after SOPs division. Note that the expression of *neur > RFP* is restricted to the four cells of the bristles. (**b**) Diagram of the tactile bristle lineage showing the cellular events characterizing the specification of the four cells of each tactile bristle. Circles indicate cells and lines indicate the duration of each event in hAPF. The cross over the glial cell indicates apoptosis. (**c**) Time course of early bristle development showing the asymmetric cell divisions and intra-lineage

apoptosis leading to the four cells of each bristle. The SOP and progeny cells are marked by the expression of *neur > RFP* and the membrane marker *pon::GFP* (grey), which displays asymmetric localization during cell divisions. (**d**) Shape of the tactile bristle by the onset of its terminal differentiation and coloured schematic showing the four component cells. Note that the socket and shaft cells have increased nuclear size, and that the neuron and sheath cells lie adjacent to each other. (**e**) Early differentiating tactile bristles marked by the expression of *neur > RFP* and the ubiquitous microtubule marker *Jupiter::GFP* (green) showing hair shaft outgrowth soon after the four bristle cells have been specified. Results are representative of three independent experiments. Scale bars: 5 μm. Full genotypes are listed in Supplementary Table 1.

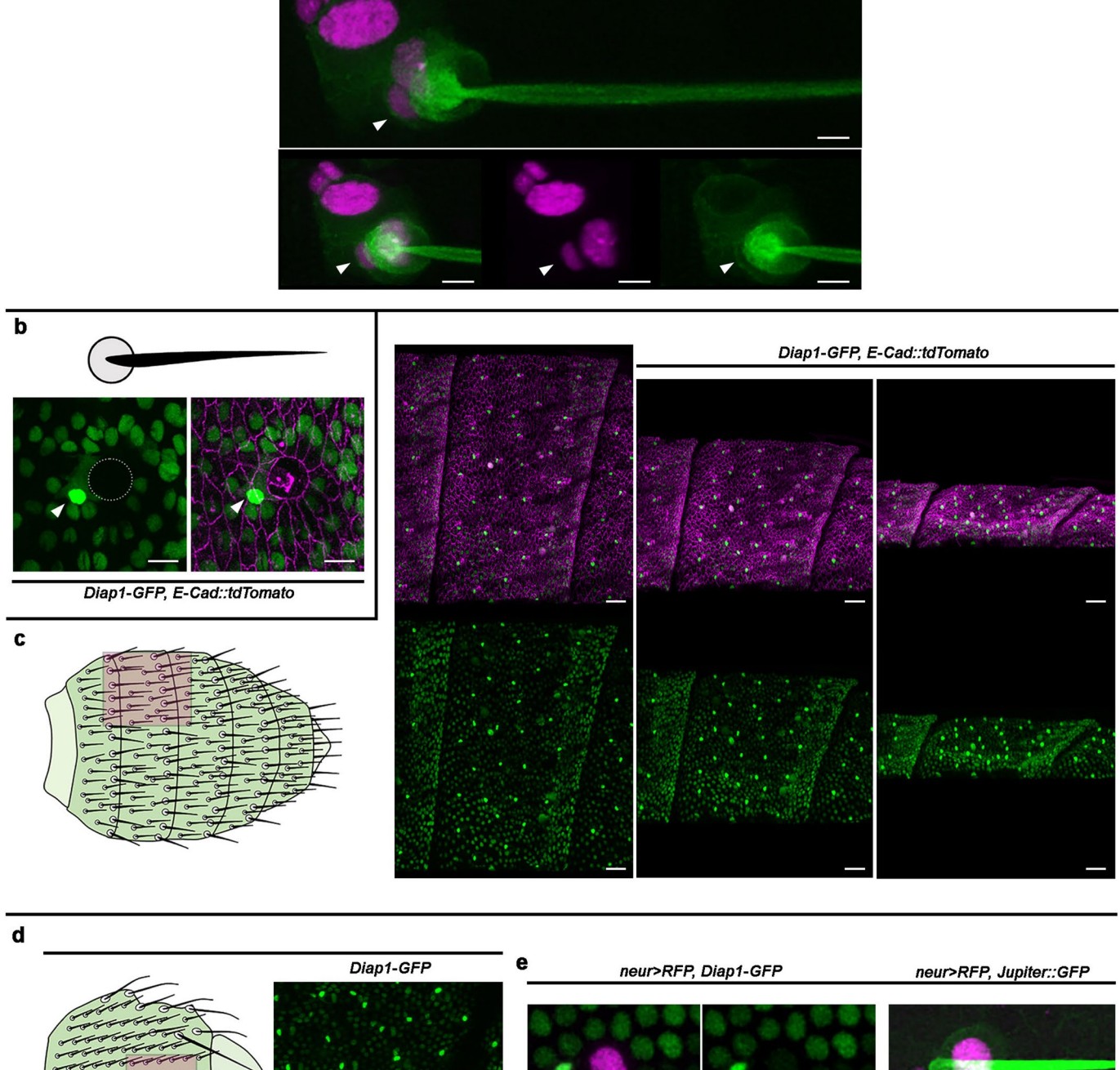

**Extended Data Fig. 2 | F-Cells lie adjacent to differentiating tactile bristles.** (**a**) Tactile bristle and the F-Cell (arrowhead) at 70 hAPF labelled by expression of the microtubule marker *GFP::CLIP-170* (green) under control of *neur > RFP*. Note that the F-Cell associates with the socket cell. (**b**) Top: diagram of the cuticular socket and hair shaft. Bottom: detection of the F-Cell (arrowhead) next to the socket cell (dotted circle) by enhanced expression of *Diap1-GFP* (green) and co-labelling of the junctional network with *E-Cad::TdTomato*. (**c**) Diagram of the tactile bristles decorating the adult abdominal surface and the abdominal epidermis visualized at 65 hAPF. *E-Cad::td-Tomato* expression marks the junctional network and *Diap1-GFP* (green) labels the nuclei of the epidermal cells. F-Cells are identifiable by elevated *Diap1-GFP* in the epidermal cells next to each socket cell within the monolayered epithelium. (**e**) Diagram of the tactile bristles decorating the adult thoracic surface and pattern of *Diap1-GFP* in the thoracic epidermis at 50 hAPF. (**d**) Identification of the F-Cell (arrowhead) in the thoracic epidermis by the expression of *Diap1-GFP* and *neur > RFP*, and simultaneous expression of *Jupiter::GFP* (green) and *neur > RFP* (magenta) to highlight F-Cell position next to the socket and shaft cells. Results are representative of three independent experiments. Scale bars: 5 μm (**a-b, e**), 15 μm (**c**) and 20 μm (**d**). Full genotypes are listed in Supplementary Table 1.

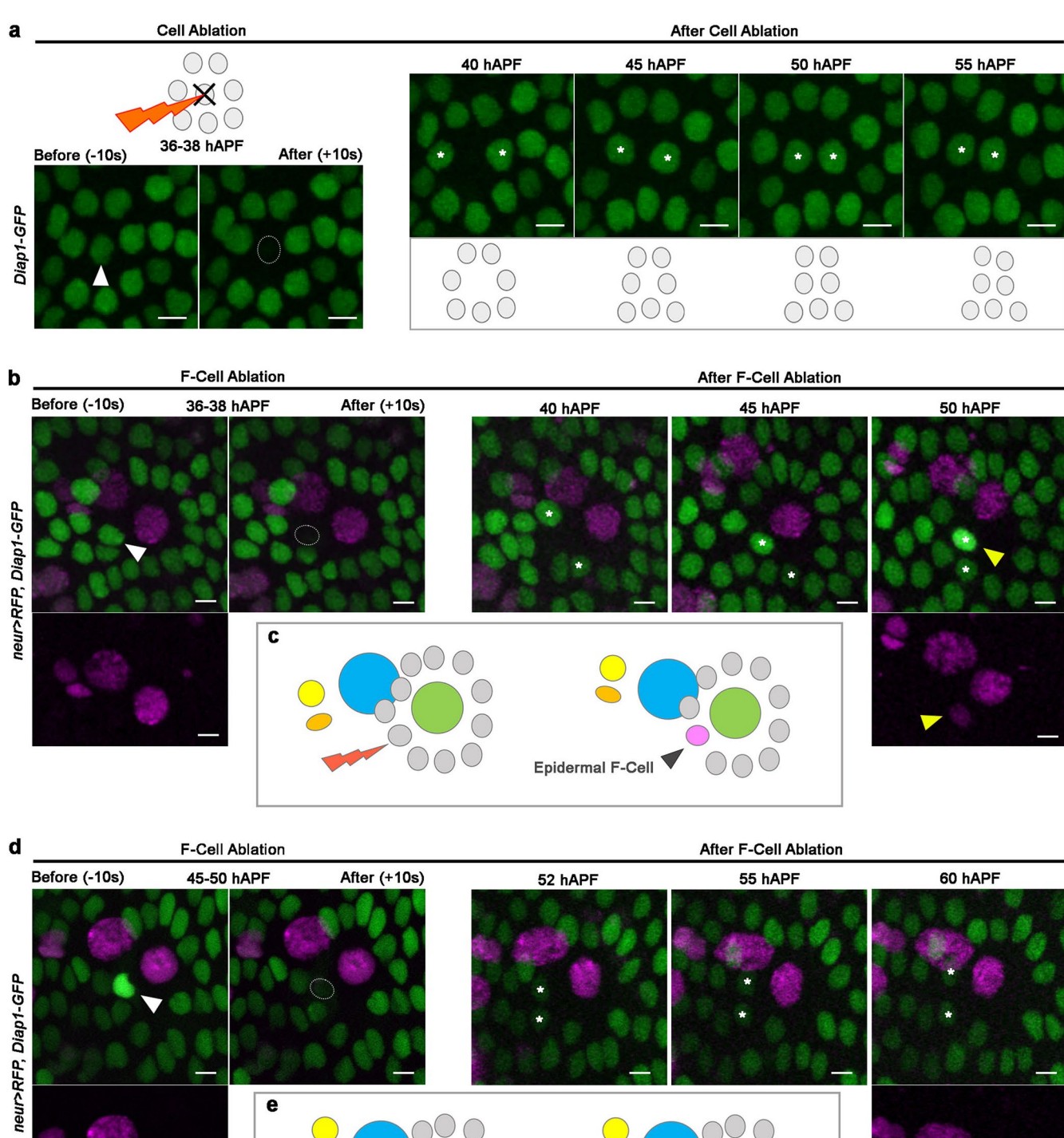

**Extended Data Fig. 3 | See next page for caption.**

**Extended Data Fig. 3 | Selective ablation of epidermal cells reveals F-Cell specification dynamics.** (**a**) Methodology used for the ablation of individual cells within the abdominal epidermis. Epidermal cells are labelled by the expression of the nuclear marker *Diap1-GFP* and depicted in grey in all diagrams. A target epidermal cell (arrowhead) is ablated using a high-power femtosecond laser pulse (see Methods), and the behaviour of surrounding cells is tracked over time. After ablations the cells adjacent to the ablated area (asterisks) move towards each other. Note that no changes in levels of expression of the *Diap1-GFP* reporter in epidermal cells are detectable over time. (**b**) Epidermal cells (*Diap1-GFP*, green) and tactile bristle cells (*neur > RFP*) before and after laser ablation of the F-Cell (white arrowhead). When the F-Cell is ablated at 36-38 hAPF, the neighbouring epidermal cells (asterisks) fill its position over time and expression levels of *Diap1-GFP* and *neur > RFP* are selectively enhanced in a single epidermal cell next to the bristle (yellow arrowheads), indicating *de novo* F-Cell specification (n = 22 F-Cells over 7 pupae from 3 independent experiments). See Supplementary Video 2. (**c**) Diagram summarizing the findings in (**b**). (**d**) Ablation of the F-Cell (arrowhead) as in (**b**) but performed at 45-50 hAPF. When the F-Cell is ablated after expressing *neur > RFP*, the cells adjacent to the ablated area (asterisks) do not show *de novo* expression of *neur > RFP* or changes in the levels of *Diap1-GFP* expression over time (n = 27 F-Cells over 9 pupae from three independent experiments. (**e**) Diagram summarizing the findings in (**d**). Results are representative of three independent experiments. Scale bars: 5 μm. Full genotypes are listed in Supplementary Table 1.

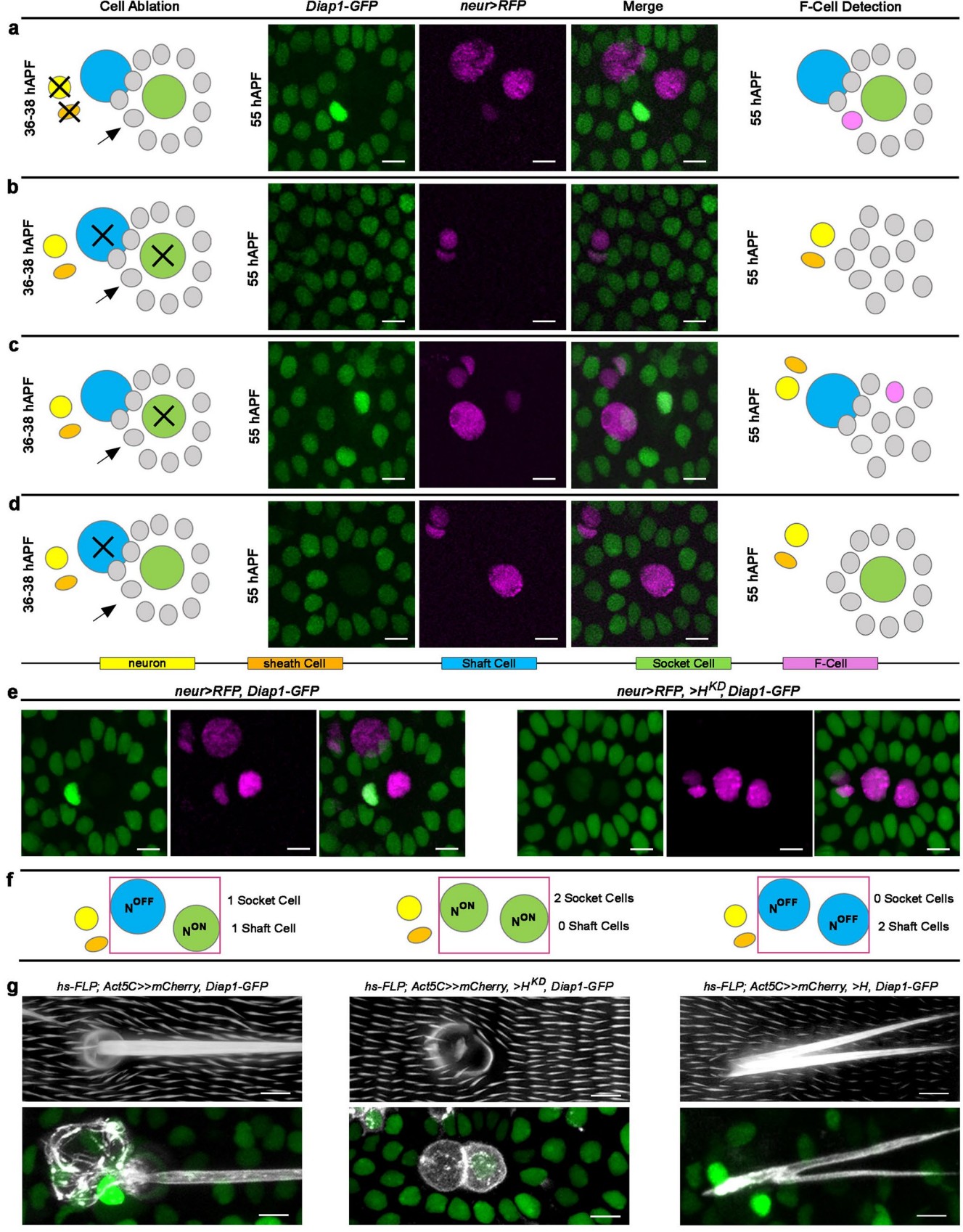

Extended Data Fig. 4 | See next page for caption.

**Extended Data Fig. 4 | F-Cell specification is initiated by the tactile bristle.** (**a**-to-**d**) Left: diagrams depicting the cell ablations used to assess F-Cell recruitment by tactile bristles cells. The cells that were ablated at 36-38 hAPF are marked by black crosses. Middle: epidermal and tactile bristle cells visualized at 55 hAPF by the expression of *Diap1-GFP* (green) and *neur > RFP*. Right: diagrams summarizing the effects of each cellular ablation on F-Cell specification (n = 16 socket cells over 7 pupae; n = 11 shaft cells over 8 pupae; n = 12 neuron and sheath cells over 5 pupae; n = 13 socket and shaft cells over 4 pupae, from three independent experiments. The F-Cell is no longer detectable after simultaneous ablation of the socket and shaft cells (**b**) or ablation of the shaft cell alone (**d**). (**e**) Effect of N activity gain via downregulation of *H* on *Diap1-GFP* and *neur > RFP* expression patterns (*neur > H^{KD}*, right) within the bristle compared to control (left). (**f**) Diagrams depicting the differential N signalling within the differentiating socket and shaft cells (green and light blue) in controls (left), upon N activity gain in the shaft cell (middle) or upon N activity loss in the socket cell (right). (**g**) Top, left-to-right: brightfield images displaying the adult cuticular socket and hair shafts in controls, two socket-like structures upon N activity gain in the shaft cell, and two hair shaft-like structures upon N activity loss in the socket cell. Bottom, left-to-right: socket and shaft cells visualized by the expression of *mCherry* in the *Diap1-GFP* background (green) showing associated F-Cells in controls, lack of F-Cell next to a bristle composed of two socket-like cells, or enhanced *Diap1-GFP* in two cells next to a bristle composed of two shaft-like cells. Results are representative of three independent experiments. Scale bars: 5 µm. Full genotypes are listed in Supplementary Table 1.

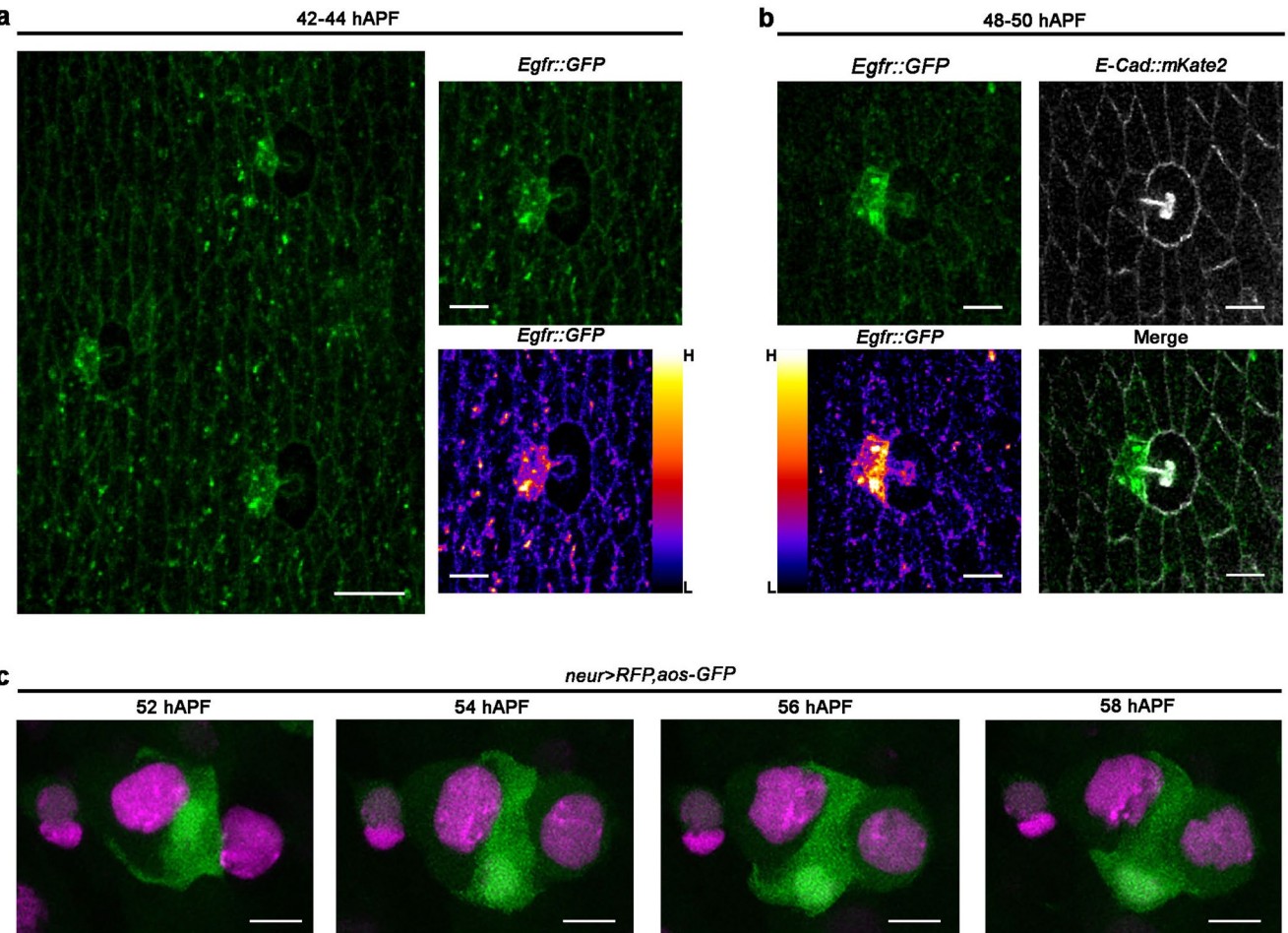

**Extended Data Fig. 5 | Elevated levels of EGFR and of the *aos-GFP* reporter in the F-Cell. (a)** EGFR protein localization during F-Cell specification. A fire LUT scale for fluorescent intensities (bottom-right) was applied to highlight higher EGFR protein levels in the F-Cell relative to surrounding epidermis. **(b)** Image showing EGFR enrichment in the F-Cell relative to epidermal cells, co-labelled by the expression of *E-Cadherin*. A fire LUT scale for fluorescent intensities (bottom-left) was applied to highlight higher protein levels of EGFR in the F-Cell relative to surrounding epidermis. **(c)** Expression pattern dynamics of the EGFR signalling reporter *aos-GFP* (green) together with *neur > RFP* during tactile bristle differentiation, revealing EGFR activity in the F-Cell. Results are representative of three independent experiments. Scale bars: 12 μm (**a**) and 5 μm (**b-c**). Full genotypes are listed in Supplementary Table 1.

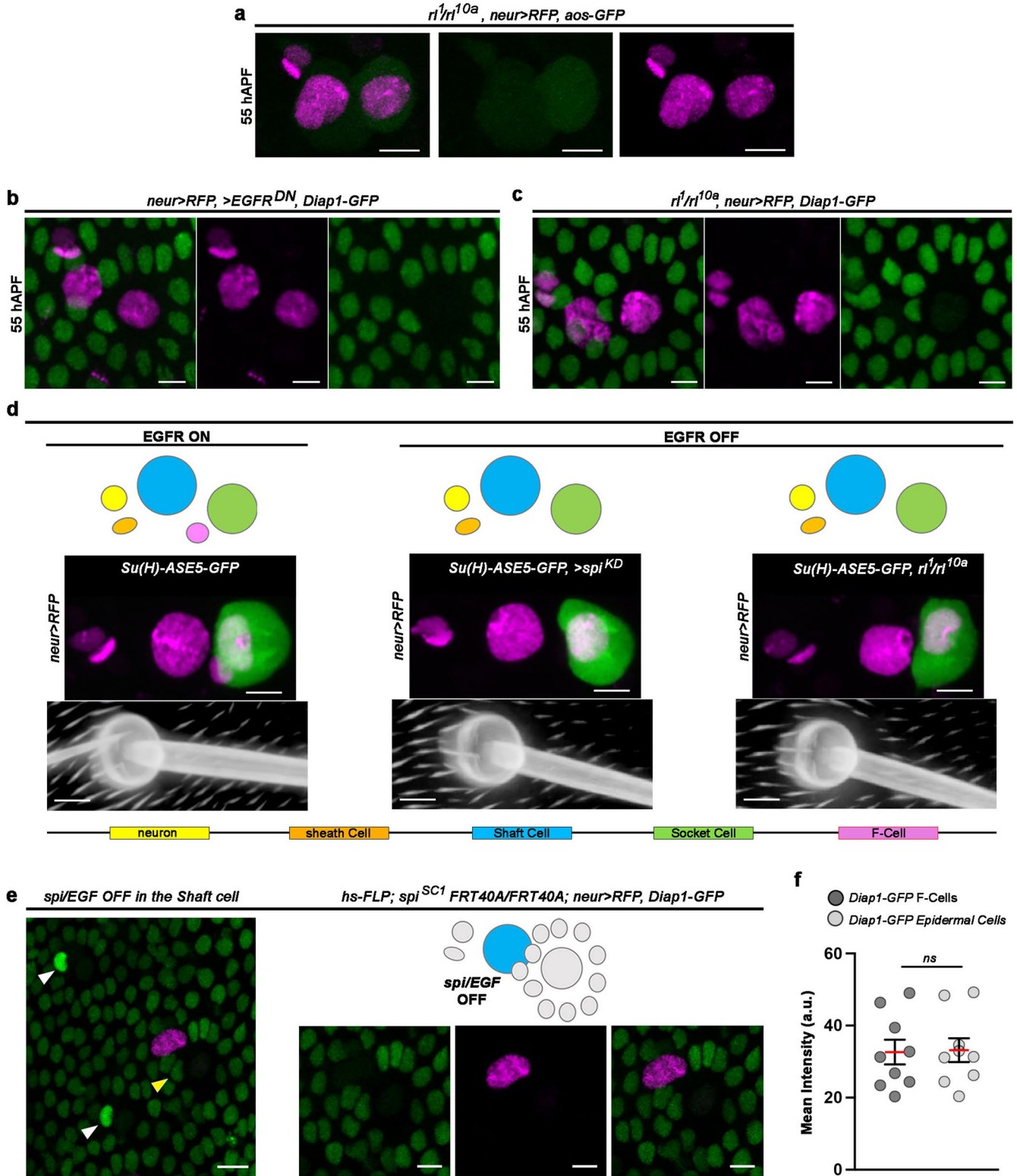

**Extended Data Fig. 6 | See next page for caption.**

**Extended Data Fig. 6 | EGFR signalling is required for F-Cell specification.**
(**a**) Expression the EGFR signalling reporter *aos-GFP* (green), together with
*neur > RFP*, in *rl/ERK* mutant background (*rl¹/rl¹⁰ᵃ*). (**b**) Uniform *Diap1-GFP*
and absence of *neur > RFP* expression in epidermal cells surrounding bristles
expressing of EGFR^DN. See Fig. 3e for quantifications. (**c**) Uniform *Diap1-GFP* and
absence of *neur > RFP* expression in epidermal cells surrounding bristles in a *rl/
ERK* mutant background. See Fig. 3e for quantifications. (**d**) Top row: diagram
of the tactile bristle when EGFR signalling is active (presence of the F-Cell) or
impaired (absence of the F-Cell). Middle row: expression of *neur > RFP* and of
the socket cell-specific reporter *Su(H)-ASE5-GFP* (green) at 60 hAPF when EGFR
signalling is active or impaired. Bottom row: bright field images showing the
shape of the cuticular socket and hair shaft at 90 hAPF in each condition. Note
that the cuticular sockets and hair shafts appear unaffected by impaired EGFR

signalling. (**e**) Right, *Diap1-GFP* expression in the epidermis showing F-Cells
adjacent to control bristles (white arrowheads), but not next to bristles lacking
*spi/EGF* activity in the shaft cells (yellow arrowhead; See Methods). Centre:
zoomed-in view of the bristle lacking *spi/EGF* activity in the shaft cells from the
left panel, with diagram summarising the result on top. (**f**) Quantification of
*Diap1-GFP* fluorescence intensity in F-Cells and epidermal cells next to shaft cells
lacking *spi/EGF* (n = 9 clones per genotypes in 8 flies at pupal stage). In the dot
plots the mean is marked in red and error bars represent SEM. The unpaired two-
tailed Student's *t*-test for equal mean was applied (p > 0.05, NS: not significant).
Results are representative of three independent experiments. Scale bars: 5 μm.
Full genotypes for are listed in Supplementary Table 1. Numerical data and exact
*P* values are available in source data.

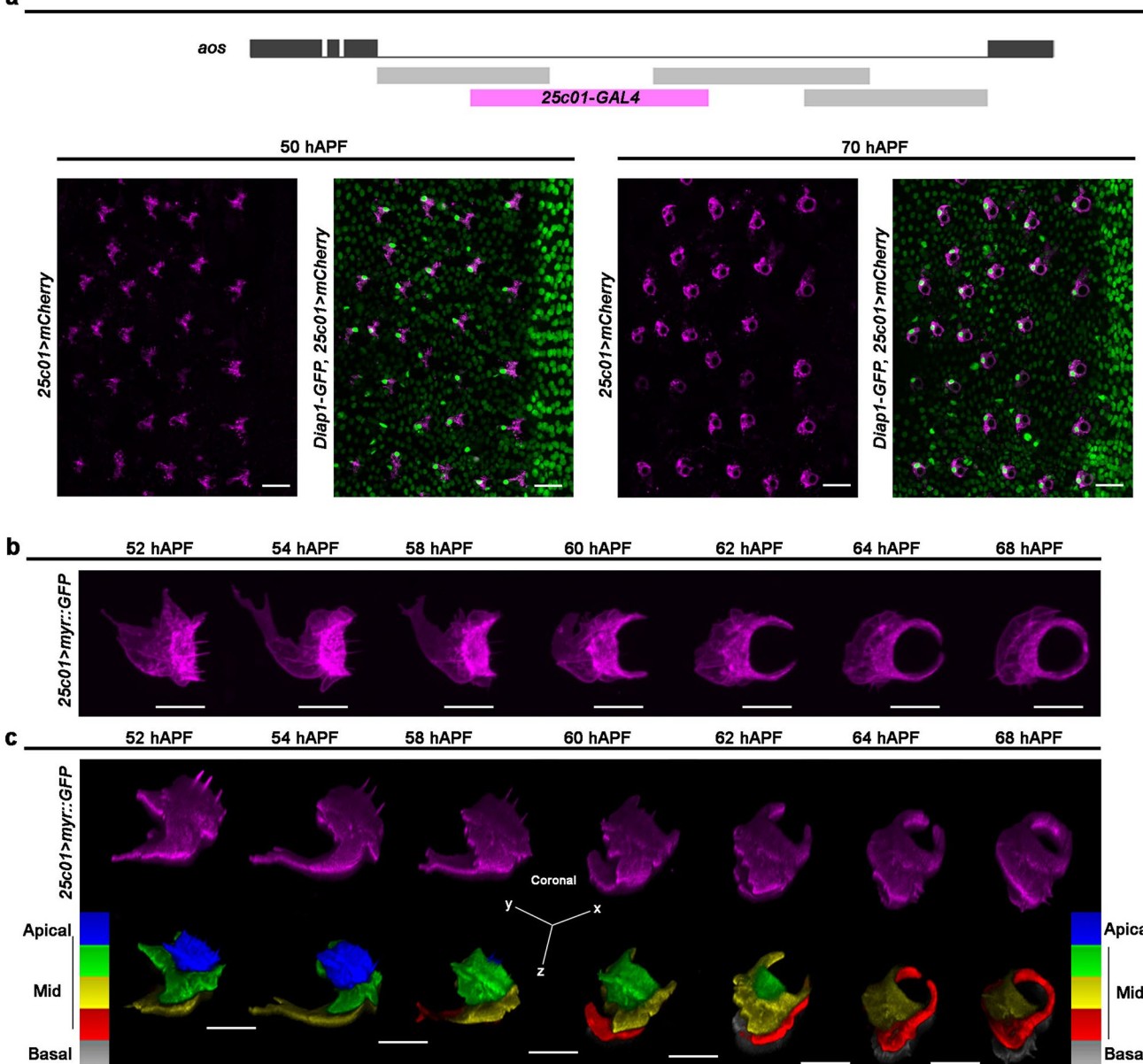

**Extended Data Fig. 7 | Morphology of differentiating F-Cells. (a)** Top: diagram of the *argos (aos)* locus. Black boxes indicate exons and black lines indicate introns. Boxes below the large intron indicate the enhancer-driven GAL4 lines screened in this study. The box highlighted in magenta denotes the GAL4 line showing restricted expression in the F-Cell. Bottom: expression of *mCherry* under the control of *25c01-GAL4* in a *Diap1-GFP* background (green) showing the co-labelling of the *mCherry* with the cells expressing high levels of *Diap1-GFP* (that is, F-Cells). **(b)** Time course of the shape changes underlying F-Cell differentiation. The morphology of the F-Cell is marked by the expression of a membrane localized GFP under the control of *25c01-GAL4* (magenta). See Supplementary Video 3. **(c)** Volume rendering of the differentiating F-Cell (coronal view) coloured in magenta (top) or after applying a 5-rumps LUT scale for z-depth (bottom). Note the basal shifting of the F-Cell as the cuticle is deposited apically over time. Results are representative of three independent experiments. Scale bar is 15 μm (**a**) and 5 μm (**b-c**). Full genotypes are listed in Supplementary Table 1.

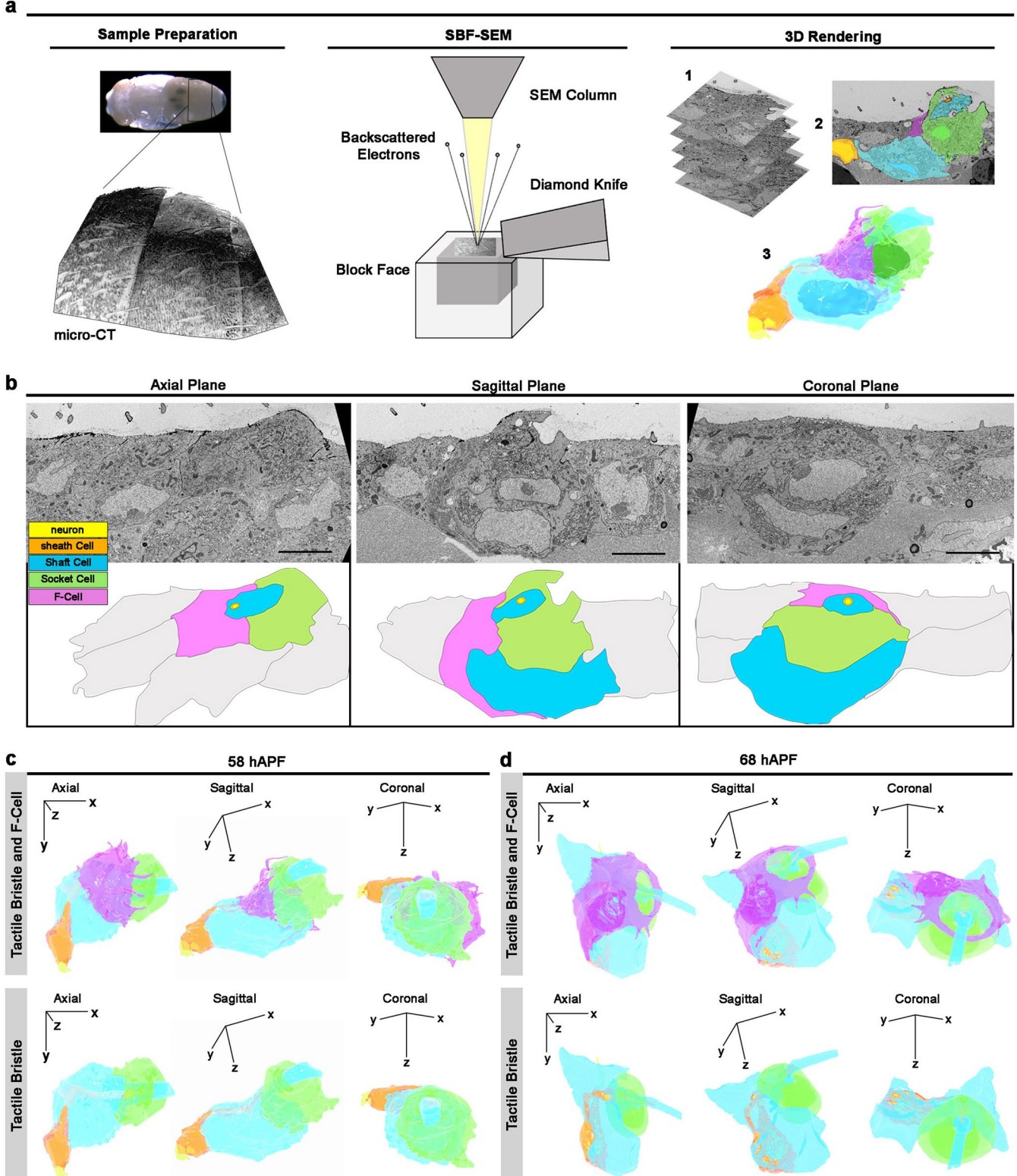

**Extended Data Fig. 8 | See next page for caption.**

**Extended Data Fig. 8 | 3D rendering of the tactile bristle and the F-Cell.**
(**a**) Workflow used to determine the 3D structure of the tactile bristle and associated F-Cell at different developmental times by Serial Block-Face Scanning Electron Microscopy (SBF-SEM). After dissection and fixation of the abdominal tissue, the embedded specimen was subjected to micro-CT (left) to orient the block face in the microscope (middle), where the specimen was sectioned and imaged. For the 3D rendering (right) multiple sections were aligned (1), the cells were manually segmented (2) and the volume within the 3D-(xyz)-space (3) was reconstructed (see Methods and Supplementary Table 2). (**b**) SBF-SEM images showing the epidermal tissue and differentiating bristle at 58 hAPF sectioned at different angles and cartoons showing the cells of the tactile bristle, the F-Cell and the epidermal cells. Note that the F-Cell is contacting both the socket and the shaft cell apically, while extending contacts toward the base of the tactile organ. (**c**) Axial (x-y), sagittal (y-z), and coronal (x-z) views of the tactile bristle and F-Cell at 58 hAPF (top) or of the tactile bristle alone (bottom). The neuron and sheath cell are concentrically enwrapped apically by the shaft and socket cells while the F-Cell is located anteriorly (see Supplementary Videos 4 and 5). (**d**) Axial, sagittal, and coronal views of the tactile bristle and F-Cell (top row), or of the tactile bristle alone (bottom row) at 68 hAPF. The neuron and sheath cell are concentrically enwrapped by the shaft cell, socket cell, and F-Cell by this stage (see Supplementary Video 6). Results are representative of three independent experiments. Full genotypes are listed in Supplementary Table 1.

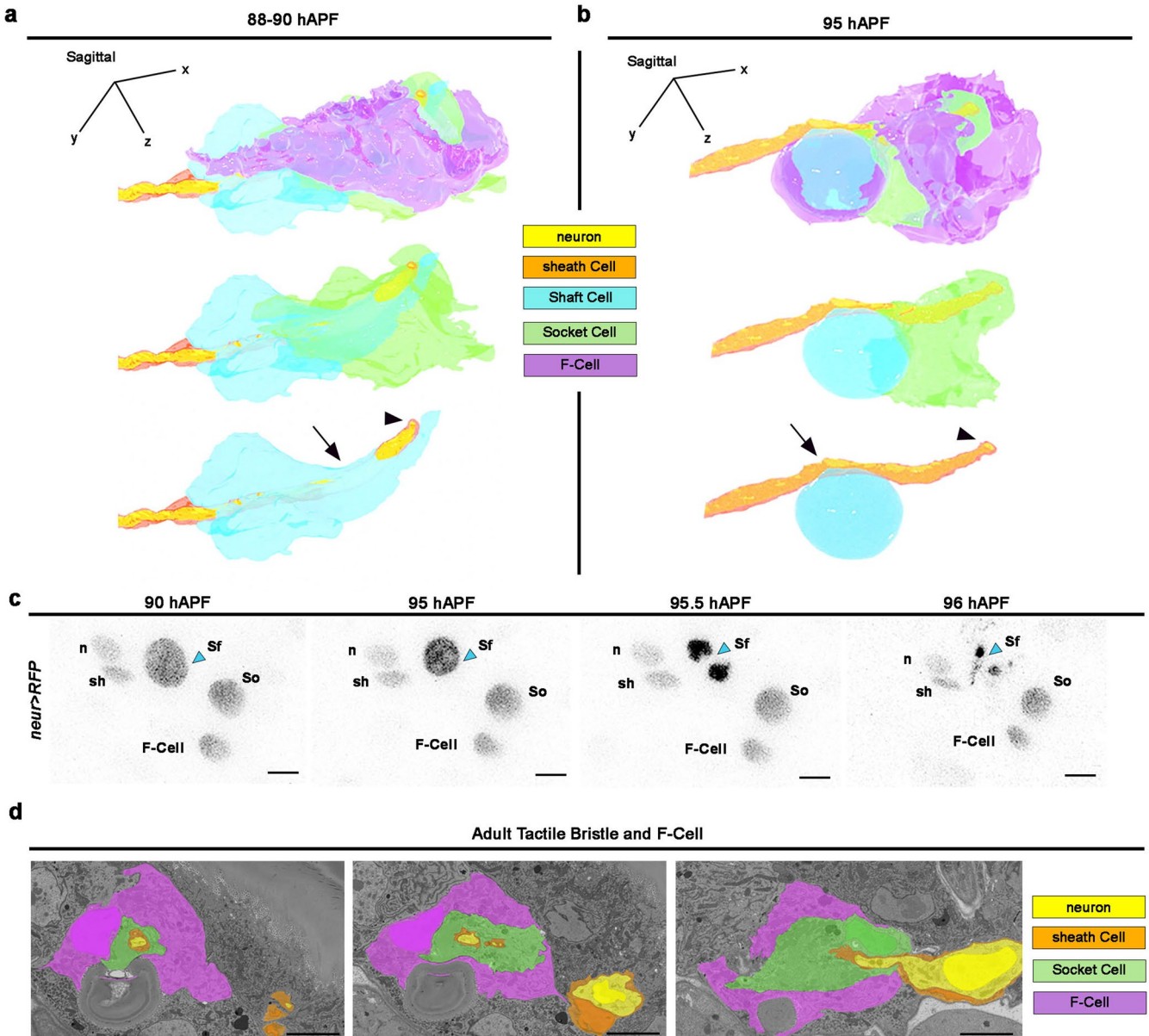

**Extended Data Fig. 9 | The F-Cell is associated with the tactile bristle of the adult fly.** (**a**) 3D morphology of the tactile bristle at 88-90 hAPF. The F-Cell shows extensive contacts with the socket cell and reduced contact with the shaft cell at this stage. The shaft cell cytoplasm (arrow) is retracting from the dendrite tip (arrowhead). See Supplementary Video 7. (**b**) 3D morphology of the tactile organ at 95 hAPF (that is, immediately before eclosion of the adult fly). The F-Cell shows extensive contacts with the socket cell and reduced contact with the shaft cell at this stage. The shaft cell cytoplasm (arrow) has retracted further away from the dendrite tip (arrowhead) and is displaying an apoptotic morphology. See

Supplementary Video 8. (**c**) Time course of the death of the shaft cell. Cells are marked by the expression of *neur > RFP*. Inverted time-lapse confocal images showing condensation and fragmentation of the shaft cell nucleus (arrowhead). Note that the F-Cell is detectable next to the socket cell also at 96 hAPF. (**d**) SBF-SEM images of the adult bristle. Note that the F-Cell is ensheathing the socket cell. Results are representative of two (**a,b**) and three (**c,d**) independent experiments. See Supplementary Video 9. Scale bar is 5 μm. Full genotypes are listed in Supplementary Table 1.

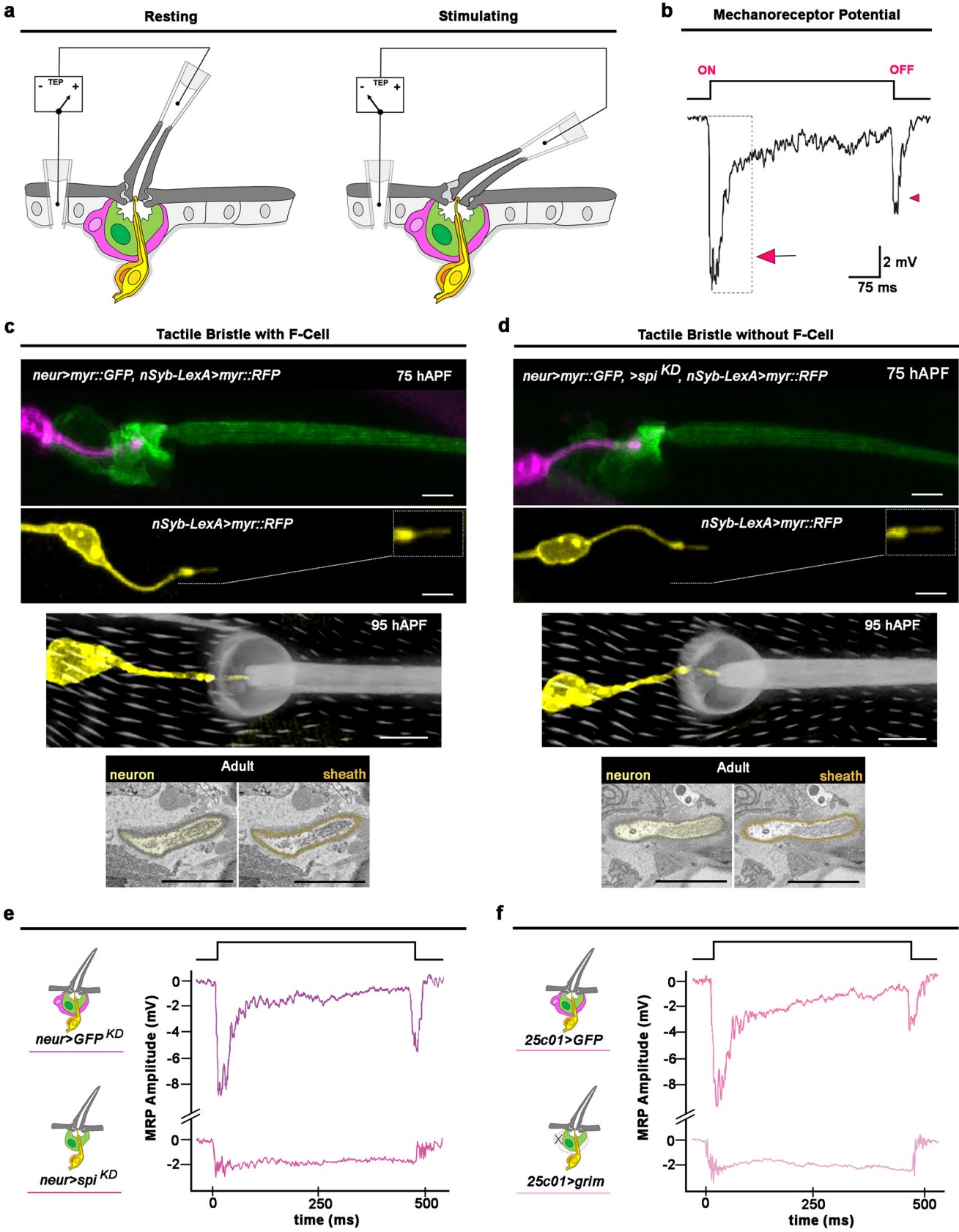

**Extended Data Fig. 10 | See next page for caption.**

**Extended Data Fig. 10 | Assessing touch-evoked mechanotransduction in the tactile bristle.** (**a**) Diagram showing the set-up for extracellular recording from tactile bristles. The three non-neuronal cells of the tactile organ (F-Cell, magenta; socket cell, green; sheath cell, orange), the mechanosensory neuron (yellow) and its connection to the base of the hair shaft are shown. Clipping the hollow hair shaft, placing a recording electrode over the tip and a reference electrode in the supporting epithelium allows the measurement of the transepithelial potential (TEP) at rest and provides electrical access to the underlying neuron. Displacement of the hair shaft toward the body evokes a robust downward drop in the TEP, measured as a mechanoreceptor potential (MRP) of the neuron (see Methods). (**b**) MRP and superimposed action potential trains recorded upon 30 μm ramp-and-hold displacement of the hair shaft toward the body surface. The neuron generates a robust response at the onset of the stimulus (arrow). The MRP slowly declines towards resting values during the stimulus. A robust but smaller response is also recorded when the neuron returned to its resting position (arrowhead). (**c-d**) Morphology and innervation in controls (**c**) or in tactile bristles lacking the F-Cell (**d**). Top row: morphology and innervation of the bristle. Second row: morphology of the mechanosensory neuron, with dendritic tip zoomed-in view displayed in top right corner. Third row: dendritic tip insertion at the base of the bristle. Bottom row: SBF-SEM section of the dendritic tip. Note that both the structure and innervation of the tactile bristle, as well as the morphology of the adult neuron and sheath cell are normal when F-Cell specification is prevented. (**e**) Representative voltage traces from control bristles (top) and bristles lacking the F-Cell (bottom). See Fig. 6 for quantifications. (**f**) Representative traces from control bristles (top) and bristles with ablated F-Cell (bottom). See Fig. 6 for quantifications. Results are representative of three independent experiments. Scale bars: 5 μm. Full genotypes are listed in Supplementary Table 1.

# Reporting Summary

## Statistics

For all statistical analyses, confirm that the following items are present in the figure legend, table legend, main text, or Methods section.

| n/a | Confirmed | |
|---|---|---|
| ☐ | ☒ | The exact sample size (*n*) for each experimental group/condition, given as a discrete number and unit of measurement |
| ☒ | ☐ | A statement on whether measurements were taken from distinct samples or whether the same sample was measured repeatedly |
| ☐ | ☒ | The statistical test(s) used AND whether they are one- or two-sided<br>*Only common tests should be described solely by name; describe more complex techniques in the Methods section.* |
| ☒ | ☐ | A description of all covariates tested |
| ☐ | ☒ | A description of any assumptions or corrections, such as tests of normality and adjustment for multiple comparisons |
| ☐ | ☒ | A full description of the statistical parameters including central tendency (e.g. means) or other basic estimates (e.g. regression coefficient) AND variation (e.g. standard deviation) or associated estimates of uncertainty (e.g. confidence intervals) |
| ☐ | ☒ | For null hypothesis testing, the test statistic (e.g. *F*, *t*, *r*) with confidence intervals, effect sizes, degrees of freedom and *P* value noted<br>*Give P values as exact values whenever suitable.* |
| ☒ | ☐ | For Bayesian analysis, information on the choice of priors and Markov chain Monte Carlo settings |
| ☒ | ☐ | For hierarchical and complex designs, identification of the appropriate level for tests and full reporting of outcomes |
| ☒ | ☐ | Estimates of effect sizes (e.g. Cohen's *d*, Pearson's *r*), indicating how they were calculated |

*Our web collection on statistics for biologists contains articles on many of the points above.*

## Software and code

Policy information about availability of computer code

| Data collection | For electrophysiology recordings, data were collected with MultiClamp700B amplifier and Axon Digidata 1550A A/D board (Molecular Devices). Microscopy data (Laser Scanning Microscopy and Electron Microscopy) were acquired using the Zen software (Zeiss LSM 880, LSM780 and Sigma VP SEM, ZEN 2.1). SEM images were acquired with a JEOL Microscope (JCM-6000Plus). |
|---|---|
| Data analysis | GraphPad Prism 9 was used for data representation and the PAST 4.04 software was used for statistical analyses. Images were analyzed using Fiji and image segmentation was performed using the TrakEM2 plugin of Fiji. Electrophysiology recordings were analyzed off line with the Clampfit software v10.7 (Molecular Devices). Animations of Volume EM data were made with Blender (software v2,9). Adobe Photoshop 2022 and Adobe Illustrator 2022 were used for figure assembly and diagrams. No custom data analysis code was required in this study. |

For manuscripts utilizing custom algorithms or software that are central to the research but not yet described in published literature, software must be made available to editors and reviewers. We strongly encourage code deposition in a community repository (e.g. GitHub). See the Nature Portfolio guidelines for submitting code & software for further information.

## Data

Policy information about availability of data

All manuscripts must include a data availability statement. This statement should provide the following information, where applicable:
- Accession codes, unique identifiers, or web links for publicly available datasets
- A description of any restrictions on data availability
- For clinical datasets or third party data, please ensure that the statement adheres to our policy

The data supporting the findings of this study are available within this paper and associated Extended Data, Source Data and Supplementary Information. Additional information are available from the corresponding authors on reasonable request.

# Field-specific reporting

Please select the one below that is the best fit for your research. If you are not sure, read the appropriate sections before making your selection.

☒ Life sciences ☐ Behavioural & social sciences ☐ Ecological, evolutionary & environmental sciences

For a reference copy of the document with all sections, see nature.com/documents/nr-reporting-summary-flat.pdf

# Life sciences study design

All studies must disclose on these points even when the disclosure is negative.

| Sample size | No statistical methods were used to predetermine sample size.  Sample sizes were chosen on the basis of previous experience, published literature and current standards in the field (PMID: 30517870, PMID: 10744543, PMID: 27807168, PMID: 12441297). Statistical analyses were performed comparing different groups (e.g., mutants versus controls) with similar sample size. Test for normal distribution were performed before choosing the statistical test for significance to be applied in any given dataset. |
|---|---|
| Data exclusions | No data points/outliers were excluded from the analyses. |
| Replication | Experiments were typically replicated at least three times, unless stated otherwise, and only those experiments for which repeats gave comparable outcomes were included in the manuscript. |
| Randomization | Samples were allocated to experimental groups according to genotype, and appropriate controls were performed in all experiments, so randomization was not required. Adult females were used as control and experimental samples for all behavioural assays and for electrophysiology. |
| Blinding | The investigator were not blinded to allocation during experiments or analysis, as group allocation was clearly visible in samples due to phenotypic and/or behavioural changes. |

# Reporting for specific materials, systems and methods

We require information from authors about some types of materials, experimental systems and methods used in many studies. Here, indicate whether each material, system or method listed is relevant to your study. If you are not sure if a list item applies to your research, read the appropriate section before selecting a response.

## Materials & experimental systems

| n/a | Involved in the study |
|---|---|
| ☒ | ☐ Antibodies |
| ☒ | ☐ Eukaryotic cell lines |
| ☒ | ☐ Palaeontology and archaeology |
| ☐ | ☒ Animals and other organisms |
| ☒ | ☐ Human research participants |
| ☒ | ☐ Clinical data |
| ☒ | ☐ Dual use research of concern |

## Methods

| n/a | Involved in the study |
|---|---|
| ☒ | ☐ ChIP-seq |
| ☒ | ☐ Flow cytometry |
| ☒ | ☐ MRI-based neuroimaging |

## Animals and other organisms

Policy information about studies involving animals; ARRIVE guidelines recommended for reporting animal research

| Laboratory animals | Fruit flies (Drosophila melanogaster) at pupal stages (males and females) and at adult stage (females). |
|---|---|
| Wild animals | No wild animals were used in this study. |
| Field-collected samples | This study did not involved samples collected from the field. |
| Ethics oversight | No ethical approval was required. The use of Drosophila melanogaster does not require ethical approval or guidance. |

Note that full information on the approval of the study protocol must also be provided in the manuscript.

