## [Peer Review File · Nature Cell Biology]

Peer Review Information

Journal: Nature Cell Biology

Manuscript Title: Co-option of epidermal cells enables touch sensing

Corresponding author name(s): Dr Nicolas Tapon

Editorial Notes:

Reviewer Comments & Decisions:

Decision Letter, initial version:
--

*Please delete the link to your author homepage if you wish to forward this email to co-authors.

Dear Dr Tapon,

Your manuscript, "Co-option of epidermal cells for touch sensing", has now been seen by 3 referees, who are experts in *Drosophila* neurobiology (referee 1); *Drosophila* mechanobiology (referee 2); and touch mechanosensation (referee 3). As you will see from their comments (attached below) they find this work of potential interest, but have raised substantial concerns, which in our view would need to be addressed with considerable revisions before we can consider publication in Nature Cell Biology.

Nature Cell Biology editors discuss the referee reports in detail within the editorial team, including the chief editor, to identify key referee points that should be addressed with priority, and requests that

are overruled as being beyond the scope of the current study. To guide the scope of the revisions, I have listed these points below. We are committed to providing a fair and constructive peer-review process, so please feel free to contact me if you would like to discuss any of the referee comments further.

In particular, it would be essential to:

A) Perform experiments to further delineate the molecular mechanisms of differentiation of F-cells and action of EGF and Diap1 (Reviewers #1 and #2)

B) Assess whether F-Cells can influence sensory neuronal firing, whether direct measurements of sensory neurons (Reviewer #3) or otherwise.

C) Quantify effects (all Reviewers) as well as reword for clarity and accessibility (Reviewer #3)

D) All other referee concerns pertaining to strengthening existing data, providing controls, methodological details, clarifications and textual changes, should also be addressed.

E) Finally please pay close attention to our guidelines on statistical and methodological reporting (listed below) as failure to do so may delay the reconsideration of the revised manuscript. In particular please provide:

We would be happy to consider a revised manuscript that would satisfactorily address these points, unless a similar paper is published elsewhere, or is accepted for publication in Nature Cell Biology in the meantime.

- ensure that it conforms to our format instructions and publication policies (see below and <https://www.nature.com/nature/for-authors>).

- provide a point-by-point rebuttal to the full referee reports verbatim, as provided at the end of this letter.

- provide the completed Reporting Summary (found here <https://www.nature.com/documents/nr-reporting-summary.pdf>). This is essential for reconsideration of the manuscript will be available to editors and referees in the event of peer review. For more information see <http://www.nature.com/authors/policies/availability.html> or contact me.

When submitting the revised version of your manuscript, please pay close attention to our [Digital Image Integrity Guidelines](https://www.nature.com/nature-portfolio/editorial-policies/image-integrity). and to the following points below:

Nature Cell Biology is committed to improving transparency in authorship. As part of our efforts in this direction, we are now requesting that all authors identified as 'corresponding author' on published papers create and link their Open Researcher and Contributor Identifier (ORCID) with their account on the Manuscript Tracking System (MTS), prior to acceptance. ORCID helps the scientific community achieve unambiguous attribution of all scholarly contributions. You can create and link your ORCID from the home page of the MTS by clicking on 'Modify my Springer Nature account'. For more information please visit www.springernature.com/orcid.

This journal strongly supports public availability of data. Please place the data used in your paper into a public data repository, or alternatively, present the data as Supplementary Information. If data can only be shared on request, please explain why in your Data Availability Statement, and also in the correspondence with your editor. Please note that for some data types, deposition in a public repository is mandatory - more information on our data deposition policies and available repositories appears below.

[Redacted]

We would like to receive a revised submission within six months.

We hope that you will find our referees' comments, and editorial guidance helpful. Please do not hesitate to contact me if there is anything you would like to discuss.

Best wishes,

Daryl Jason David

Daryl J.V. David, PhD

Senior Editor, Nature Cell Biology
Consulting Editor, Nature Communications
Nature Portfolio

Heidelberger Platz 3, 14197 Berlin, Germany
Email: daryl.david@nature.com
ORCID: <https://orcid.org/0000-0002-9253-4805>

Reviewers' Comments:

Reviewer #1:

Remarks to the Author:

In this manuscript, Mangione et al has described the developmental origin and property of the fifth cell in SOP development using live imaging and SEM combined with genetic perturbations. They have shown that the fifth cell is important for touch-sensing. The subject is interesting, and the experiments are nicely presented.

Specific comments:

1. The authors found that EGFR pathway is required for the F-cell differentiation by downregulating the expression of EGFR ligand spi and reducing the EGFR downstream signaling using ERK/MAPK loss of function allele. However, the authors should examine whether the protein level of EGFR in F-cells is different from other epidermal cells and whether manipulating the expression of EGFR in F-cells (maybe using 25c01-Gal4) affects the F-cell differentiation as ERK/MAPK has other upstream signals than EGFR. It is also interesting to see whether the EGFR ligand spi is from shaft cells.
2. Any idea how an epidermal cell is singled out to be the F-cell? Perhaps experiment described in (1) may provide some clue.
3. The finding that "F-Cell specification occurs post-mitotically" is very interesting. Is shaft cell required for maintaining F-cell fate? What happens to the developing F-cell if the shaft cell is ablated after the recruited epidermal cells have already acquired F-cell properties?
4. As the expression level of the enhancer of Diap1 (Diap1-GFP) increased specifically in the F-cell, does Diap1 play any role in the F-cell differentiation?
5. In Extended data Fig. 4.C, it seems that more than one epidermal cell showed increased Diap1-GFP expression by ablation of socket cell. Did the authors find this consistently?
6. The authors found that tactile bristles that lack F-cells show reduced mechano-transduction, does lack of F-cells affect the morphology of the neuron or the expression/localization of the mechano-sensor NOMPC?

Reviewer #2:

Remarks to the Author:

The manuscript by N. Tapon and colleagues describes the organization and differentiation of mechanosensory organs of the *Drosophila* adult epidermis. They identified that an epidermal cell that they name F-Cell, differentiates from the other epidermal cells and is necessary for mechanosensation. This F-cell presents a specific expression of Diap1 compared to the remaining tissue and undergo an amazing cell shape change to surround the bristle. The presence of the shaft cell and EGF signalling are essential for the specification of the F-cell. With functional assays, the authors show that the presence and specificity of the F-cell is necessary for the mechanosensation of the fly. Overall, this is a remarkable work that opens new potential research way to understand mechanosensing in animals. The quality of the study is very high, few quantifications are missing, particularly on the timing of the different events, but with some corrections, I believe this work would be suitable to be published in Nature Cell Biology.

Specific comments:

- On the colocalization of neur>RFP and diap1-GFP, on the fig 2. The authors mention that "...Diap1-GFP levels increased specifically in the F-Cell, concurrently with the appearance of neur>RFP...". it is difficult to assess the temporal evolution of the two markers: From comparing fig 1 and 2 it seems that Diap1GFP is enriched at earlier time points. A quantification of fluorescence intensity over time of the two markers will allow the reader to assess the temporal evolution of the two markers. It would be also at later stage in the manuscript to relate the kinetics of shape changes with the levels of neur>RFP and Diap1 GFP.
- The specific enrichment of Diap1 in the F-cell is very surprising. What happen if Diap1 is downregulated in this cell? Or in the case of cell ablation in which several cells get Diap 1 enrichment, will they differentiate? The author could discuss the potential role/function of Diap1 in the process.
- From the fig 2, the enrichment in Diap 1 is associated to a change in nuclear shape from spherical to "crescent shape". This change in nuclear shape seems to be consequent to the morphological changes of the F-cell and is present in many figures. It is absent when the shaft cell is ablated (extends fig4-c) but is present in EGFR off (fig 3 D). Would this suggests that EGFR affect differentiation but not cell shape changes?
- The F-Cell undergoes a remarkable cell shape change to surround the bristle. This dramatic change in shape suggests a differential adhesion between the F-cell and bristle vs the other cells. How this shape change is dependent on adhesion molecules, are cadherin or integrin important for this process to happen?

Reviewer #3:

Remarks to the Author:

A. Summary of the key results

This study deciphers the developmental, anatomical and functional properties of F-Cells in the *drosophila* bristle complex and their contributions to touch sensation. The authors start by describing the anatomical features of F-cells and their associations with bristles on the dorsal body surface of

drosophila, suggesting that F-cells associate with differentiating tactile bristles in different body parts. The authors also describe (via very convincing laser ablation studies) that F-cell specification occurs post-mitotically. They continue to show that both bristles and epidermal cells engage with specific F-cells through Notch signaling. By manipulating EGFR signaling mechanisms, the authors also show that it is required for F-cell recruitment into bristles. In a set of very elegant morphological studies (including time-lapse imaging and serial block-face SEM), they show how F-Cells associate with bristle. These morphological studies allow the authors to make discrete hypotheses as to their function in bristle mediated mechanotransduction, which they continue to show in another set of elegant studies measuring transepithelial potential. With these sets of studies they show that the on/off potentials are reduced in F-Cell ablated and genetic mutant flies. Taking a step further the authors show that eliminating f-cells does have a consequence in touch-induced reflexes.

Overall this is a quite impressive study! Taking it from the identification of the cells, to their development and function. While this is done in a model organism that is not within my field of expertise, I believe that this is a very important study for the field of touch and mechanotransduction. Highlighting the contribution of non-neuronal cell-types to electrophysiological signatures of touch sensitive sensory neurons. The overall conclusions of this study also provide very clear and testable hypotheses as to how F-Cells contribute to these on/off responses.

B. Originality and significance: if not novel, please include reference.

I do not have a lot of experience or background on the use of drosophila as a model organism. However, based on the references that the authors included, a full characterization of F-Cells in bristle mechanotransduction is quite novel. For the touch/mechanotransduction field their findings are significant as they provide fundamental insights into how non-neuronal cells sculpt the activity of sensory neurons. Within my immediate field of expertise this study aligns itself (at least in significance and originality) to this study: Maksimovic et al. Nature 2014 (PMID: 24717432), in which the authors show how the merkel cell contributes to the slowly-adapting properties of Ab-LTMRs that innervate touch domes in hairy skin.

C. Data & methodology: validity of approach, quality of data, quality of presentation

Validity of approach and quality of data: their approaches are outside of my immediate field of expertise but based on my knowledge the authors use established methodologies throughout. The femtosecond laser ablation method seems to work quite well. Based on the reference that they provide in the methodology (#58), this study seems to be the first time that this particular methodology is used in drosophila (?). If so, this can be quite novel. However, I wonder: heat damage to neighboring cells?

Quality of presentation: The data is quite beautiful, including the images (some are simply stunning!). However, for a non-expert this study was hard to get through. The schematic "dots" (see Fig. 1b) can be confusing since this is a 3-dimensional structure. I found that the schematics in Fig.5a is much more effective and I found myself referring to that quite often to make sense of the data. Possible to use that schematic throughout OR have it in figure 1 and strategically placed in other figures as needed for clarity? Also for the non-drosophila researchers, the figures can get quite jargonny (hAPF, neur>RFP, etc). Possible to simplify the headings used in all of the figures? The figure legend can be a place where drosophila aficionados can identify particular crosses and gestation periods. Also, there seems to be a missed opportunity for clarity in having the figures in the panels match that of the

schematic throughout (for example: as much as possible F-cells should be one color throughout).

D. Appropriate use of statistics and treatment of uncertainties

This study is largely descriptive with ablation and genetic studies showing one example. However, the figure legend lacks some basic information as to the number of animals/cells analyzed. Perhaps the figures and the study as a whole could benefit from some quantitation of the phenomena that the authors are observing. For example, in Figure 2b: how many cells were ablated, what percentage of them resulted in de novo specification? While some of this information can be found in the methodology (see "single cell ablation with two-photon microscopy") having this information in the figure legend can be helpful, as it was hard for me to decipher which ablation experiment described in the methodology corresponded to which figure. For the electrophysiology and behavior this is largely not the case as individual points are shown (for ephys, though not for behavior, Fig.5g)

E. Conclusions: robustness, validity, reliability

The conclusions that the authors propose are in line with their experimental observations. Depending on how widely accepted the targeted laser ablation and the extracellular recordings are (the latter established in 1994? ref#39 Kernan et al. Neuron 1994) then their results support the conclusions that the authors make.

F. Suggested improvements: experiments, data for possible revision

Again, not my field of expertise, but if direct sensory neuron recording is possible, we might learn a bit more about how F-Cell influences the firing properties of sensory neurons. I also understand that this might not be possible with current techniques.

G. References: appropriate credit to previous work?

Based on the type of foundational work that this study accomplished and how this manuscript is written, as a reader it seems that up until this study F-Cells have not been widely characterized (anatomically or otherwise). While this may be true, it is often the case that some of the anatomy (maybe basic cell types?), have been characterized and/or identified by someone before. In the field that I am most familiar with, most of the foundational anatomical work was pioneered by mammalian anatomists in the 1800s and 1900s. Our work tends to cite these anatomists, both for appropriate recognition but also to highlight the significance of our work. However, it might just be the case that this IS the first time F-Cells have been described. If the latter, then this study also offers a lot of novelty on top of significance.

H. Clarity and context: lucidity of abstract/summary, appropriateness of abstract, introduction and conclusions.

Clarity and context is adequate, however please see comments above regarding the quality of the presentation.

FINANCIAL AND NON-FINANCIAL COMPETING INTERESTS – the authors must include one of three declarations: (1) that they have no financial and non-financial competing interests; (2) that they have financial and non-financial competing interests; or (3) that they decline to respond, after the Author Contributions section. This statement will be published with the article, and in cases where financial and non-financial competing interests are declared, these will be itemized in a web supplement to the article. For further details please see <https://www.nature.com/licenceforms/nrg/competing->

interests.pdf.

Methods should be written concisely, but should contain all elements necessary to allow interpretation and replication of the results. As a guideline, Methods sections typically do not exceed 3,000 words. The Methods should be divided into subsections listing reagents and techniques. When citing previous methods, accurate references should be provided and any alterations should be noted. Information must be provided about: antibody dilutions, company names, catalogue numbers and clone numbers for monoclonal antibodies; sequences of RNAi and cDNA probes/primers or company names and catalogue numbers if reagents are commercial; cell line names, sources and information on cell line identity and authentication. Animal studies and experiments involving human subjects must be reported in detail, identifying the committees approving the protocols. For studies involving human subjects/samples, a statement must be included confirming that informed consent was obtained. Statistical analyses and information on the reproducibility of experimental results should be provided in a section titled "Statistics and Reproducibility".

All Nature Cell Biology manuscripts submitted on or after March 21 2016 must include a Data availability statement as a separate section after Methods but before references, under the heading "Data Availability". For Springer Nature policies on data availability see <http://www.nature.com/authors/policies/availability.html>; for more information on this particular policy see <http://www.nature.com/authors/policies/data/data-availability-statements-data-citations.pdf>. The Data availability statement should include:

- Accession codes for primary datasets (generated during the study under consideration and designated as "primary accessions") and secondary datasets (published datasets reanalysed during the study under consideration, designated as "referenced accessions"). For primary accessions data should be made public to coincide with publication of the manuscript. A list of data types for which submission to community-endorsed public repositories is mandated (including sequence, structure, microarray, deep sequencing data) can be found here <http://www.nature.com/authors/policies/availability.html#data>.
- Unique identifiers (accession codes, DOIs or other unique persistent identifier) and hyperlinks for datasets deposited in an approved repository, but for which data deposition is not mandated (see here for details <http://www.nature.com/sdata/data-policies/repositories>).

- At a minimum, please include a statement confirming that all relevant data are available from the authors, and/or are included with the manuscript (e.g. as source data or supplementary information), listing which data are included (e.g. by figure panels and data types) and mentioning any restrictions on availability.
- If a dataset has a Digital Object Identifier (DOI) as its unique identifier, we strongly encourage including this in the Reference list and citing the dataset in the Methods.

We recommend that you upload the step-by-step protocols used in this manuscript to the Protocol Exchange. More details can found at www.nature.com/protocolexchange/about.

All imaging data should be accompanied by scale bars, which should be defined in the legend. Cropped images of gels/blots are acceptable, but need to be accompanied by size markers, and to retain visible background signal within the linear range (i.e. should not be saturated). The boundaries of panels with low background have to be demarked with black lines. Splicing of panels should only be considered if unavoidable, and must be clearly marked on the figure, and noted in the legend with a statement on whether the samples were obtained and processed simultaneously. Quantitative comparisons between samples on different gels/blots are discouraged; if this is unavoidable, it should only be performed for samples derived from the same experiment with gels/blots were processed in parallel, which needs to be stated in the legend.

- We accept PowerPoint (.PPT) files if they are fully editable. However, please refrain from adding PowerPoint graphical effects to objects, as this results in them outputting poor quality raster art. Text used for PowerPoint figures should be Helvetica (preferred) or Arial.
- We do not recommend using Adobe Photoshop for designing figures, but we can accept Photoshop generated (.PSD or .TIFF) files only if each element included in the figure (text, labels, pictures, graphs, arrows and scale bars) are on separate layers. All text should be editable in 'type layers' and line-art such as graphs and other simple schematics should be preserved and embedded within 'vector smart objects' - not flattened raster/bitmap graphics.
- Some programs can generate Postscript by 'printing to file' (found in the Print dialogue). If using an application not listed above, save the file in PostScript format or email our Art Editor, Allen Beattie for advice (a.beattie@nature.com).

The total number of Supplementary Figures (not including the “unprocessed scans” Supplementary Figure) should not exceed the number of main display items (figures and/or tables (see our Guide to Authors and March 2012 editorial <http://www.nature.com/ncb/authors/submit/index.html#suppinfo>; <http://www.nature.com/ncb/journal/v14/n3/index.html#ed>). No restrictions apply to Supplementary Tables or Videos, but we advise authors to be selective in including supplemental data.

GUIDELINES FOR EXPERIMENTAL AND STATISTICAL REPORTING

REPORTING REQUIREMENTS – We are trying to improve the quality of methods and statistics reporting in our papers. To that end, we are now asking authors to complete a reporting summary that collects information on experimental design and reagents. The Reporting Summary can be found here <https://www.nature.com/documents/nr-reporting-summary.pdf>) If you would like to reference the guidance text as you complete the template, please access these flattened versions at <http://www.nature.com/authors/policies/availability.html>.

We strongly recommend the presentation of source data for graphical and statistical analyses as a separate Supplementary Table, and request that source data for all independent repeats are provided

when representative experiments of multiple independent repeats, or averages of two independent experiments are presented. This supplementary table should be in Excel format, with data for different figures provided as different sheets within a single Excel file. It should be labelled and numbered as one of the supplementary tables, titled "Statistics Source Data", and mentioned in all relevant figure legends.

Author Rebuttal to Initial comments

We thank all three reviewers for their valuable comments, which have helped us improve the manuscript. Here is our detailed response. Reviewers' comments are transcribed verbatim in black italic font. Our responses are in **blue** font.

Reviewer #1:

Remark to the Author:

In this manuscript, Mangione et al has described the developmental origin and property of the fifth cell in SOP development using live imaging and SEM combined with genetic perturbations. They have shown that the fifth cell is important for touch-sensing. The subject is interesting, and the experiments are nicely presented.

Specific comments:

1. The authors found that EGFR pathway is required for the F-cell differentiation by downregulating the expression of EGFR ligand spi and reducing the EGFR downstream signaling using ERK/MAPK loss of function allele. However, the authors should examine whether the protein level of EGFR in F-cells is different from other epidermal cells and whether manipulating the expression of EGFR in F-cells (maybe using 25c01-Gal4) affects the F-cell differentiation as ERK/MAPK has other upstream signals than EGFR. It is also interesting to see whether the EGFR ligand spi is from shaft cells.

As requested, we have now performed several experiments to elucidate the role of EGFR in F-Cell specification.

- We examined EGFR localization using the *EGFR::sfGFP* transgene recently described in (Revaitis et al., 2020). This analysis revealed that EGFR protein levels are indeed higher in the F-Cell during its specification. These data are now shown in Extended Data Figure 5a-b.

- We investigated whether impaired activity of EGFR itself impacts F-Cell specification. Expression of a dominant negative form of EGFR (*EGFR-DN*, FBst0005364) in *neur*-expressing cells (Bristles and F-Cells) was sufficient to abrogate F-Cell specification, as we had previously shown to be the case for the EGFR ligand *spi*. These results are now shown in Extended Data Figure 6b. We next blocked EGFR activity in the F-Cell at the time of its specification (FLP-Out method for marked mosaics; see revised Methods for details) and found that this was sufficient to abolish F-Cell fate specification. These data are now shown in Figure 3f-g. Note that, in these experiments, we used the FLP-Out method since the expression of *25c01-GAL4* is activated after F-Cell fate has been established (Extended Data Fig.7c), therefore the *EGFR-DN* construct would be expressed too late to interfere with EGFR activation.

- We investigated whether the Shaft cell acts as a source of the EGFR ligand Spi. To address this with a high degree of precision, we generated marked mutant clones for *spi* (MARCM clones; see revised Methods for details). We tested for the presence of the F-Cell in bristles where the Shaft cell was mutant for *spi*. We found that F-Cell specification was abolished, indicating that expression of *spi* in the Shaft cell is required to initiate F-Cell specification. This result is now shown in Extended Data Figure 6e-f.

Collectively, these experiments provide further evidence that EGFR signalling is required for F-Cell specification.

2. Any idea how an epidermal cell is singled out to be the F-cell? Perhaps experiment described in (1) may provide some clue.

Our data, which now include the experiments provided in response to point 1, suggest that initiation of F-Cell specification occurs in response to the secretion of Spi from the Shaft cell of the bristle, with

which the F-Cell is in direct contact. Together with F-Cell biased EGFR expression, secreted Spi from the Shaft cell triggers restricted activation of EGFR signalling in the F-Cell but not its epithelial cell neighbours. At present, we do not know how EGFR expression is modulated in the F-Cell and future work will be needed to elucidate this and other aspects of F-Cell recruitment.

3. The finding that “F-Cell specification occurs post-mitotically” is very interesting. Is shaft cell required for maintaining F-cell fate? What happens to the developing F-cell if the shaft cell is ablated after the recruited epidermal cells have already acquired F-cell properties?

We performed laser ablation experiments targeting the Shaft cell after F-Cell specification. We found that neither the expression of the F-Cell markers (*neur>RFP* and enhanced levels of *Diap1-GFP*), nor the morphological differentiation of the F-Cell were affected (for Reviewer, Figure 1 for R1.3). Therefore, although essential for the initiation of the F-Cell fate, the Shaft cell is dispensable for F-Cell differentiation.

Figure 1 for R1.3

Figure 1 for R1.3. (a) Images showing the F-Cell (arrowhead), marked by the expression of *neur>RFP* (magenta) and enhanced expression of *Diap1-GFP* (green) at 65 hAPF. (b) Images showing the F-Cell at

65 hAPF (arrowhead) after ablation of the Shaft cell at 50 hAPF. Note that the morphological differentiation of the F-Cell has occurred normally in the absence of the Shaft cell, as seen by the faint cytoplasmic GFP signal in panel b which shows that the F-Cell has ensheathed the bristle. Scale bars: 10 μ m.

4. As the expression level of the enhancer of *Diap1* (*Diap1-GFP*) increased specifically in the F-cell, does *Diap1* play any role in the F-cell differentiation?

In this work we primarily used the *Diap1-GFP* enhancer, a 4.3 kb regulatory region of *Diap1* originally described in (Zhang et al., 2008), as a marker for F-Cell fate. We nonetheless agree that investigating the role of *Diap1* in specified F-Cells is an interesting question. Since DIAP1 has a key role as caspase inhibitor in the control of apoptosis (Vasudevan and Ryoo, 2015), one possibility is that enhanced *Diap1* expression could act as pro-survival signal during the morphological differentiation of the F-Cell. Consistent with this idea, downregulation of *Diap1* expression in the F-Cell was sufficient to induce F-Cell death (for Reviewer, Figure 2 for R1.4).

Figure 2 for R1.4

Figure 2 for R1.4. (a-b) Snapshots showing F-Cell (arrowhead) morphological changes with (a) or without (b) downregulation of *Diap1* in the F-Cell. Note extrusion of the F-Cell upon *Diap1*- downregulation. Scale bars: 10 μ m.

In some cases, however, DIAP1 can also modulate the activity of caspases fulfilling non-apoptotic functions (Colon-Plaza and Su, 2022), leaving open the possibility that DIAP1 might not merely act as pro-survival factor during F-Cell differentiation. However, given that DIAP1 depletion leads to rapid cell death, we feel that investigating this potential role would be very difficult to address experimentally and would require a separate study.

5. In Extended data Fig. 4.C, it seems that more than one epidermal cell showed increased Diap1-GFP expression by ablation of socket cell. Did the authors find this consistently?

It is true that, upon ablation of the Socket cell, more than one adjoining epidermal cell sometimes show a slight increase in *Diap1-GFP*, as in the example provided in Extended Data Fig. 4c. Nonetheless, in these experiments, we consistently observed that only one cell upregulates both *Diap1-GFP* and *neur>RFP*, a hallmark of F-Cell specification.

6. The authors found that tactile bristles that lack F-cells show reduced mechano-transduction, does lack of F-cells affect the morphology of the neuron or the expression/localization of the mechano-sensor NOMPC?

In the original submission, we had addressed whether the lack of the F-Cell has an impact on the morphology of the neuron using live imaging (Extended Data Fig. 10c,d). We had observed no defects, both in the cell body and the dendrite tip reaching the base of the bristle shaft, indicating that the F-Cell is not influencing neuronal morphology. We examined neuronal tip morphology in more detail using electron microscopy and found no obvious differences in bristles with or without F-Cells. These data are now shown in Extended Data Figure 10c-d. These observations suggest that the mechano-sensor NOMPC, which localizes at the tip of the neuron (Lee et al., 2010) is unlikely to be affected. However, in

our hands the available NOMPC antibody does not give a reliable staining that would allow use to compare the mutant and control bristles, therefore we were unable to test this directly.

Reviewer #2: (Note that we numbered the reviewer's comments in blue)

Remarks to the Author:

The manuscript by N. Tapon and colleagues describes the organization and differentiation of mechanosensory organs of the Drosophila adult epidermis. They identified that an epidermal cell that they name F-Cell, differentiates from the other epidermal cells and is necessary for mechanosensation. This F-cell presents a specific expression of Diap1 compared to the remaining tissue and undergo an amazing cell shape change to surround the bristle. The presence of the shaft cell and EGF signalling are essential for the specification of the F-cell. With functional assays, the authors show that the presence and specificity of the F-cell is necessary for the mechanosensation of the fly. Overall, this is a remarkable work that opens new potential research way to understand mechanosensing in animals. The quality of the study is very high, few quantifications are missing, particularly on the timing of the different events, but with some corrections, I believe this work would be suitable to be published in Nature Cell Biology.

Specific comments:

1. On the colocalization of $neur>RFP$ and $diap1-GFP$, on the fig 2. The authors mention that "...Diap1-GFP levels increased specifically in the F-Cell, concurrently with the appearance of $neur>RFP$...". it is difficult to assess the temporal evolution of the two markers: From comparing fig 1 and 2 it seems that Diap1GFP is enriched at earlier time points. A quantification of fluorescence intensity over time of the two markers will allow the reader to assess the temporal evolution of the two markers. It would be also at later stage in the manuscript to relate the kinetics of shape changes with the levels of $neur>RFP$ and Diap1 GFP.

As requested, we have performed quantifications of fluorescence intensities of *Diap1-GFP* and *neur>RFP* to elucidate the temporal evolution of these two markers during F-Cell specification. These quantifications revealed that *Diap1-GFP* is indeed enriched at earlier time points than *neur>RFP*.

However, the expression of both markers progressively increases in the F-Cell by 41-42 hAPF onward. These data are now shown in Figure 2b (see revised Methods for details). By 50-55 hAPF, and thus at the onset of F-Cell morphological differentiation, both *Diap1-GFP* and *neur>RFP* levels reached their maxima, being more than 3-fold higher in the F-Cells than the surrounding epidermal cells. These quantifications are now shown in Figure 2c. We have also performed quantifications of *Diap1-GFP* and *neur>RFP* levels upon manipulations of EGFR signalling and found that levels of expression of both F-Cell markers remain basal in absence of EGFR signalling. These quantifications are now shown in Figure 3e. The text has been amended to clarify this for the reader.

2. The specific enrichment of Diap1 in the F-cell is very surprising. What happens if Diap1 is downregulated in this cell? Or in the case of cell ablation in which several cells get Diap1 enrichment, will they differentiate? The author could discuss the potential role/function of Diap1 in the process.

In our manuscript, the expression of *Diap1-GFP*, a 4.3 kb regulatory region of *Diap1* originally described in (Zhang et al., 2008), was used as a marker for F-Cell fate. The enhancement of *Diap1-GFP* in the F-Cell is indeed very intriguing. This can be the result of combinatorial inputs into the regulatory region of this reporter, as we and others observed that *Diap1-GFP* expression is modulated by multiple signalling pathways including N and EGFR (Djiane et al., 2013; Reiff et al., 2019). In response to the reviewer's question on "What happens if *Diap1* is downregulated in this cell?", we have silenced the expression of *Diap1* in the F-Cells and found that F-Cells undergo cell death (for Reviewer, Figure 1 for R2.2). Considering the key function of DIAP1 as caspase inhibitor (Vasudevan and Ryoo, 2015), one possibility is that enhanced *Diap1* expression could act as pro-survival signal to prevent apoptosis of the F-Cell during bristle differentiation.

Figure 1 for R2.2
Figure 1 for R2.2. (a-b) Snapshots showing F-Cell (arrowhead) morphological changes with (a) or without (b) downregulation of *Diap1* in the F-Cell. Note extrusion of the F-Cell upon *Diap1*- downregulation. Scale bars: 10µm.

In some cases, however, DIAP1 can also modulate the activity of caspases fulfilling non-apoptotic functions (Colon-Plaza and Su, 2022), leaving open the possibility that DIAP1 might not merely act as pro-survival factor during F-Cell differentiation. However, given that DIAP1 depletion leads to rapid cell death, we feel that investigating this potential role would be very difficult to address experimentally and would require a separate study.

In response to the second point of the reviewer, which can be more broadly phrased as “Does *Diap1-GFP* enrichment result in F-Cell differentiation?”, we found that enhanced levels of *Diap1-GFP*, by themselves, are not sufficient to make an epidermal cell into an F-Cell. Our data show that a combination of signalling inputs from, and contact sharing with, a neighbouring bristle are both essential for the acquisition of F-Cell fate (e.g., both *Diap1-GFP* and *neur>RFP*).

3. From the fig 2, the enrichment in *Diap 1* is associated to a change in nuclear shape from spherical to "crescent shape". This change in nuclear shape seems to be consequent to the morphological changes of the F-cell and is present in many figures. It is absent when the shaft cell is ablated (extends fig4-c) but is present in *EGFR* off (fig 3 D). Would this suggests that *EGFR* affect differentiation but not cell shape changes?

The reviewer makes an interesting point about the "crescent" shape of the nucleus that the F-Cell acquires during bristle morphogenesis. Our observations suggest that the nuclear morphology of the F-Cell is not a direct consequence of F-Cell fate acquisition. As pointed out by the reviewer, nuclear shape appears as a "crescent" in presence or absence of *EGFR* signalling, where bristle morphology normal (Fig. 3c-to-g; Extended Data Fig. 6d and Extended Data Fig. 9c,d), while it is "spherical" when the bristle is morphologically abnormal (ablations: Extended Data Fig 4a-to-d; or altered cell fate in the bristle: Extended Data Fig 4e,g-h). We therefore believe that nuclear deformation of cells surrounding the bristles is influenced by the mechanical constraints elicited by the morphogenesis of the bristle itself. The F-cell therefore likely acquires its crescent nucleus simply because of its stereotypical position between the Shaft and Socket rather than because of its differentiation *per se*.

4. The F-Cell undergoes a remarkable cell shape change to surround the bristle. This dramatic change in shape suggests a differential adhesion between the F-cell and bristle vs the other cells. How this shape change is dependent on adhesion molecules, are cadherin or integrin important for this process to happen?

As suggested by the reviewer, we examined the expression of classical cadherins including E-cadherin and N-cadherin, whose differential expression modulate cell shape acquisition in other epithelia, including the *Drosophila* retina (Hayashi and Carthew, 2004; Chan et al., 2017). However, our investigations did not reveal detectable differences in E-Cad expression levels in the F-Cell or bristle cells, and N-Cad expression was not present in any intercellular junction (for Reviewer, Figure 2 for R2.4a,b). We also examined the expression of β -integrin (Mys, FBgn0004657) and found homogenous expression in bristles, F-Cells and epidermal cells (for Reviewer, Figure 2 for R2.4c). Future studies will

be needed to address whether F-Cell morphogenesis involves cytoskeletal remodelling and/or differential adhesion implicating other cell surface proteins.

Figure 2 for R2.4

Figure 2 for R2.4. (a-b) Images showing E-cad (magenta) and N-Cad (green) expression in the epidermis (a; F-Cells arrowhead), and in the retina epithelium (b; cone cells arrowhead). (c) Images showing E-Cad (left) and Mys (right) expression in the epidermis. The F-Cell is arrowhead. Scale bars: 10µm.

Reviewer #3: (Note that we numbered the reviewer's comments in blue)

Remarks to the Author:

A. Summary of the key results

This study deciphers the developmental, anatomical and functional properties of F-Cells in the drosophila bristle complex and their contributions to touch sensation. The authors start by describing the anatomical features of F-cells and their associations with bristles on the dorsal body surface of drosophila, suggesting that F-cells associate with differentiating tactile bristles in different body parts. The authors also describe (via very convincing laser ablation studies) that F-cell specification occurs post-mitotically. They continue to show that both bristles and epidermal cells engage with specific F-cells through Notch signaling. By manipulating EGFR signaling mechanisms, the authors also show that it is required for F-cell recruitment into bristles. In a set of very elegant morphological studies (including time-lapse imaging and serial block-face SEM), they show how F-Cells associate with bristle. These morphological studies allow the authors to make discrete hypotheses as to their function in bristle mediated mechanotransduction, which they continue to show in another set of elegant studies measuring transepithelial potential. With these sets of studies they show that the on/off potentials are reduced in F-Cell ablated and genetic mutant flies. Taking a step further the authors show that eliminating f-cells does have a consequence in touch-induced reflexes.

Overall this is a quite impressive study! Taking it from the identification of the cells, to their development and function. While this is done in a model organism that is not within my field of expertise, I believe that this is a very important study for the field of touch and mechanotransduction. Highlighting the contribution of non-neuronal cell-types to electrophysiological signatures of touch sensitive sensory neurons. The overall conclusions of this study also provide very clear and testable hypotheses as to how F-Cells contribute to these on/off responses.

B. Originality and significance: if not novel, please include reference.

I do not have a lot of experience or background on the use of drosophila as a model organism. However, based on the references that the authors included, a full characterization of F-Cells in bristle mechanotransduction is quite novel. For the touch/mechanotransduction field their findings are significant as they provide fundamental insights into how non-neuronal cells sculpt the activity of sensory neurons. Within my immediate field of expertise this study aligns itself (at least in significance and originality) to this study: Maksimovic et al. Nature 2014 (PMID: 24717432), in which the authors show how the merkel cell contributes to the slowly-adapting properties of Ab-LTMRs that innervate touch domes in hairy skin.

C. Data & methodology: validity of approach, quality of data, quality of presentation

1. Validity of approach and quality of data: their approaches are outside of my immediate field of expertise but based on my knowledge the authors use established methodologies throughout. The femtosecond laser ablation method seems to work quite well. Based on the reference that they provide in the methodology (#58), this study seems to be the first time that this particular methodology is used in drosophila (?). If so, this can be quite novel. However, I wonder: heat damage to neighboring cells?

With the targeted cell ablation methodology we have implemented in this study, the thermal damage induced by the laser pulse is confined to the cell of interest via multiphoton microscopy, which ensure spatial confinement of laser pulse. We have not observed appreciable signs of damage to neighbouring cells in our experiments, using both long-term imaging following ablation, and careful inspection of the adult cuticle.

Quality of presentation:

2. The data is quite beautiful, including the images (some are simply stunning!). However, for a non-expert this study was hard to get through. The schematic “dots” (see Fig. 1b) can be confusing since this is a 3-dimensional structure. I found that the schematics in Fig.5a is much more effective and I found

myself referring to that quite often to make sense of the data. Possible to use that schematic throughout OR have it in figure 1 and strategically placed in other figures as needed for clarity? Also for the non-drosophila researchers, the figures can get quite jargonny (hAPF, neur>RFP, etc). Possible to simplify the headings used in all of the figures? The figure legend can be a place where drosophila aficionados can identify particular crosses and gestation periods. Also, there seems to be a missed opportunity for clarity in having the figures in the panels match that of the schematic throughout (for example: as much as possible F-cells should be one color throughout).

To improve the clarity of presentation for the reader, we have revised some of our schematic diagrams throughout the manuscript. For example, as suggested, we are now ensuring that the magenta colour highlights the F-Cell only in all schematics. Where appropriate, we have added panel labels to clarify the aim of the experiment for non-specialists (e.g. EGFR manipulations throughout Fig.3). The genotypes in the figures are already simplified and we now provide the full genotypes in supplementary table 1 for specialists to refer to.

We feel that the schematics “dots” representing the arrangement of the nuclei throughout Fig. 1 to Fig. 3 (and associated Extended Data) reflects the 2D visualization of the organ/epidermal cells in the corresponding figures. It is true that the tactile bristle is a 3D structure, especially as it differentiates; hence we devoted significant effort in rendering its 3D shape by using Volume EM (Fig. 4 and associated Extended Data), which we believe it is another novel aspect of our study. However, in the early stages depicted in Figures 1-3, we believe the schematic dots are useful to help the reader navigate the corresponding images.

D. Appropriate use of statistics and treatment of uncertainties:

3. This study is largely descriptive with ablation and genetic studies showing one example. However, the figure legend lacks some basic information as to the number of animals/cells analyzed. Perhaps the figures and the study as a whole could benefit from some quantitation of the phenomena that the

authors are observing. For example, in Figure 2b: how many cells were ablated, what percentage of them resulted in *de novo* specification? While some of this information can be found in the methodology (see “single cell ablation with two-photon microscopy”) having this information in the figure legend can be helpful, as it was hard for me to decipher which ablation experiment described in the methodology corresponded to which figure. For the electrophysiology and behavior this is largely not the case as individual points are shown (for ephys, though not for behavior, Fig.5g).

As suggested by the reviewer, we provided the experimental information, including number of repeats in the figure legends. We also performed quantification of fluorescence intensity of *Diap1-GFP* and *neur>RFP* in several experiments to further support our findings (see Figure 2b, c and Figure 3e of the revised manuscript).

E. Conclusions: robustness, validity, reliability

4. The conclusions that the authors propose are in line with their experimental observations. Depending on how widely accepted the targeted laser ablation and the extracellular recordings are (the latter established in 1994? ref#39 Kernan et al. Neuron 1994) then their results support the conclusions that the authors make.

The experiments performed with the laser ablation methodology we have implemented here have been also independently validated using genetic experiments (Extended Data Fig.4). The extracellular recordings have been performed as described in (Kernan et al., 1994). This is the gold standard methodology to access the bristle neuron in *Drosophila*, adapted from earlier studies in other insects (Thurm, 1965b; Thurm and Küppers, 1980; Grunert and Gnatzy, 1987).

F. Suggested improvements: experiments, data for possible revision

5. Again, not my field of expertise, but if direct sensory neuron recording is possible, we might learn a bit more about how F-Cell influences the firing properties of sensory neurons. I also understand that this might not be possible with current techniques.

The electrophysiological recordings we performed, although extracellular, give us direct access to the firing properties of the sensory neuron, as illustrated by numerous past studies (Thurm, 1965b; Thurm and Küppers, 1980; Grunert and Gnatzy, 1987; Kernan et al., 1994; Walker et al., 2000; Dubruille et al., 2002). Upon hair shaft deflection, the recorded extracellular signal is a downward deflection of the mechanoreceptor receptor potential (MRP) and a burst of superimposed action potentials (Extended Data Fig. 10). Tactile bristles are innervated by a single bipolar neuron, therefore the action potentials superimposed on the MRP are direct extracellular measurements of sensory neuron firing (see also Walker et al., 2000). We therefore quantified both the MRP and the number of action potentials evoked in response to hair shaft deflection (Fig. 5d-f and Extended Data Figure 10b). We have amended the text to clearly communicate that our measurements directly capture sensory neuronal firing (“We next tested if F-Cells are required for neuronal firing upon touch. To address this, we quantified the action potentials superimposed on the MRP at the onset of the tactile stimulus (a direct measure of sensory neuronal firing) in control bristles or bristles lacking the F-Cell”). To our knowledge, a preparation that allows intracellular access to the sensory neuron and manipulation of its membrane potential has never been achieved for tactile bristles (also discussed in Kernan, 2007).

G. References: appropriate credit to previous work?

6. Based on the type of foundational work that this study accomplished and how this manuscript is written, as a reader it seems that up until this study F-Cells have not been widely characterized (anatomically or otherwise). While this may be true, it is often the case that some of the anatomy (maybe basic cell types?), have been characterized and/or identified by someone before. In the field that I am most familiar with, most of the foundational anatomical work was pioneered by mammalian anatomists in the 1800s and 1900s. Our work tends to cite these anatomists, both for appropriate recognition but also to highlight the significance of our work. However, it might just be the case that this

IS the first time F-Cells have been described. If the latter, then this study also offers a lot of novelty on top of significance.

The reviewer is correct in that our manuscript describes the F-Cell for the very first time. We have carefully examined previous literature in *Drosophila*, as well as in other insects, as multiple insect anatomists have been fascinated by these sensory organs (Hartenstein and Posakony, 1989; Keil, 1997; Thurm, 1965a). However, we have found neither descriptions of the epidermal cells surrounding adult tactile bristles nor examinations of their terminal differentiation. Nonetheless, we have cited past work, which include ref#6, ref#10, ref#40 and ref#41 in the original submission, because of their relevance in describing earlier steps of bristles development or their morphology. We believe that, together with the identification of the F-Cell, another novel and exciting aspect of our study is indeed the description of the terminal differentiation of the tactile bristles and its mature morphology.

H. Clarity and context: lucidity of abstract/summary, appropriateness of abstract, introduction and conclusions.

7. Clarity and context: lucidity of abstract/summary, appropriateness of abstract, introduction and conclusions. Clarity and context is adequate, however please see comments above regarding the quality of the presentation.

We believe that this point has been addressed by the above-described changes.

References:

- Chan, E.H., Chavadimane Shivakumar, P., Clement, R., Laugier, E., and Lenne, P.F. (2017). Patterned cortical tension mediated by N-cadherin controls cell geometric order in the *Drosophila* eye. *Elife* 6.
- Colon-Plaza, S., and Su, T.T. (2022). Non-Apoptotic Role of Apoptotic Caspases in the *Drosophila* Nervous System. *Front Cell Dev Biol* 10, 839358.

- Djiane, A., Krejci, A., Bernard, F., Fexova, S., Millen, K., and Bray, S.J. (2013). Dissecting the mechanisms of Notch induced hyperplasia. *EMBO J* *32*, 60-71.
- Dubruille, R., Laurencon, A., Vandaele, C., Shishido, E., Coulon-Bublex, M., Swoboda, P., Couble, P., Kernan, M., and Durand, B. (2002). Drosophila regulatory factor X is necessary for ciliated sensory neuron differentiation. *Development* *129*, 5487-5498.
- Grunert, U., and Gnatzy, W. (1987). K⁺ and Ca⁺⁺ in the receptor lymph of arthropod cuticular mechanoreceptors. *J Comp Physiol A* *161*, 329-333.
- Hartenstein, V., and Posakony, J.W. (1989). Development of adult sensilla on the wing and notum of *Drosophila melanogaster*. *Development* *107*, 389-405.
- Hayashi, T., and Carthew, R.W. (2004). Surface mechanics mediate pattern formation in the developing retina. *Nature* *431*, 647-652.
- Keil, T.A. (1997). Functional morphology of insect mechanoreceptors. *Microsc Res Tech* *39*, 506-531.
- Kernan, M., Cowan, D., and Zuker, C. (1994). Genetic dissection of mechanosensory transduction: mechanoreception-defective mutations of *Drosophila*. *Neuron* *12*, 1195-1206.
- Kernan, M.J. (2007). Mechanotransduction and auditory transduction in *Drosophila*. *Pflugers Arch* *454*, 703-720.
- Lee, J., Moon, S., Cha, Y., and Chung, Y.D. (2010). *Drosophila* TRPN(=NOMPC) channel localizes to the distal end of mechanosensory cilia. *PLoS One* *5*, e11012.
- Reiff, T., Antonello, Z.A., Ballesta-Illan, E., Mira, L., Sala, S., Navarro, M., Martinez, L.M., and Dominguez, M. (2019). Notch and EGFR regulate apoptosis in progenitor cells to ensure gut homeostasis in *Drosophila*. *EMBO J* *38*, e101346.
- Revaitis, N.T., Niepielko, M.G., Marmion, R.A., Klein, E.A., Piccoli, B., and Yakoby, N. (2020). Quantitative analyses of EGFR localization and trafficking dynamics in the follicular epithelium. *Development* *147*.
- Thurm, U. (1965a). An insect mechanoreceptor. I. Fine structure and adequate stimulus. *Cold Spring Harb Symp Quant Biol* *30*, 75-82.
- Thurm, U. (1965b). An insect mechanoreceptor. II. Receptor potentials. *Cold Spring Harb Symp Quant Biol* *30*, 83-94.
- Thurm, U., and Küppers, J. (1980). EPITHELIAL PHYSIOLOGY OF INSECT SENSILLA.
- Vasudevan, D., and Ryoo, H.D. (2015). Regulation of Cell Death by IAPs and Their Antagonists. *Curr Top Dev Biol* *114*, 185-208.
- Walker, R.G., Willingham, A.T., and Zuker, C.S. (2000). A *Drosophila* mechanosensory transduction channel. *Science* *287*, 2229-2234.
- Zhang, L., Ren, F., Zhang, Q., Chen, Y., Wang, B., and Jiang, J. (2008). The TEAD/TEF family of transcription factor Scalloped mediates Hippo signaling in organ size control. *Dev Cell* *14*, 377-387.

Decision Letter, first revision:

Our ref: NCB-A48065A

13th December 2022

Dear Dr. Tapon,

Thank you for submitting your revised manuscript "Co-option of epidermal cells for touch sensing" (NCB-A48065A). It has now been seen by the original referees and their comments are below. The reviewers find that the paper has improved in revision, and therefore we'll be happy in principle to publish it in Nature Cell Biology, pending minor revisions to satisfy the referees' final requests and to comply with our editorial and formatting guidelines.

Thank you again for your interest in Nature Cell Biology Please do not hesitate to contact me if you have any questions.

Sincerely,
Daryl

Daryl J.V. David, PhD

Senior Editor, Nature Cell Biology
Consulting Editor, Nature Communications
Nature Portfolio

Heidelberger Platz 3, 14197 Berlin, Germany
Email: daryl.david@nature.com
ORCID: <https://orcid.org/0000-0002-9253-4805>

Reviewer #1 (Remarks to the Author):

The authors have done a nice job addressed my comments and revise the manuscript. I support the publication of this paper in NCB

Reviewer #2 (Remarks to the Author):

The manuscript by Mangione et al. describes the organization and differentiation of mechanosensory organs of the *Drosophila* adult epidermis. After this first round of revision the manuscript has been improved by providing quantifications and answering concerns of the reviewers. Particularly, the shape changes of the F cell are impressive and it is surprising that none of the N or E cadherin or integrins have specific enrichment. It would have been a nice add on to hint on potential molecular mechanisms. However, I agree with the reviewers that it is going beyond the scope of this study. Overall, the study is impressive, very complete and can be published as it is in Nature Cell Biology.

Reviewer #3 (Remarks to the Author):

The authors of this study have appropriately addressed my concerns. No further comments.

Decision Letter, final checks:

Our ref: NCB-A48065A

10th January 2023

Dear Dr. Tapon,

Thank you for your patience as we've prepared the guidelines for final submission of your Nature Cell Biology manuscript, "Co-option of epidermal cells for touch sensing" (NCB-A48065A). Please carefully follow the step-by-step instructions provided in the attached file, and add a response in each row of the table to indicate the changes that you have made. Please also check and comment on any additional marked-up edits we have proposed within the text. Ensuring that each point is addressed will help to ensure that your revised manuscript can be swiftly handed over to our production team.

In recognition of the time and expertise our reviewers provide to Nature Cell Biology's editorial process, we would like to formally acknowledge their contribution to the external peer review of your

manuscript entitled "Co-option of epidermal cells for touch sensing". For those reviewers who give their assent, we will be publishing their names alongside the published article.

Nature Cell Biology offers a Transparent Peer Review option for new original research manuscripts submitted after December 1st, 2019. As part of this initiative, we encourage our authors to support increased transparency into the peer review process by agreeing to have the reviewer comments, author rebuttal letters, and editorial decision letters published as a Supplementary item. When you submit your final files please clearly state in your cover letter whether or not you would like to participate in this initiative. Please note that failure to state your preference will result in delays in accepting your manuscript for publication.

Cover suggestions

As you prepare your final files we encourage you to consider whether you have any images or illustrations that may be appropriate for use on the cover of Nature Cell Biology.

Nature Cell Biology has now transitioned to a unified Rights Collection system which will allow our Author Services team to quickly and easily collect the rights and permissions required to publish your work. Approximately 10 days after your paper is formally accepted, you will receive an email in providing you with a link to complete the grant of rights. If your paper is eligible for Open Access, our Author Services team will also be in touch regarding any additional information that may be required to arrange payment for your article.

Please note that *Nature Cell Biology* is a Transformative Journal (TJ). Authors may publish their research with us through the traditional subscription access route or make their paper immediately open access through payment of an article-processing charge (APC). Authors will not be required to make a final decision about access to their article until it has been accepted. Find out more about Transformative Journals

Authors may need to take specific actions to achieve compliance with funder and institutional open access mandates. If your research is supported by a funder that requires immediate open access (e.g. according to Plan S principles) then you should select the gold OA route, and we will direct you to the compliant route where possible. For authors selecting the subscription publication route, the journal's standard licensing terms will need to be accepted, including self-

archiving policies. Those licensing terms will supersede any other terms that the author or any third party may assert apply to any version of the manuscript.

Please use the following link for uploading these materials:
[Redated]

Best regards,

Kendra Donahue
Staff
Nature Cell Biology

On behalf of

Daryl J.V. David, PhD

Senior Editor, Nature Cell Biology
Consulting Editor, Nature Communications
Nature Portfolio

Heidelberger Platz 3, 14197 Berlin, Germany
Email: daryl.david@nature.com
ORCID: <https://orcid.org/0000-0002-9253-4805>

Reviewer #1:

Remarks to the Author:

The authors have done a nice job addressed my comments and revise the manuscript. I support the publication of this paper in NCB

Reviewer #2:

Remarks to the Author:

The manuscript by Mangione et al. describes the organization and differentiation of mechanosensory organs of the *Drosophila* adult epidermis. After this first round of revision the manuscript has been improved by providing quantifications and answering concerns of the reviewers. Particularly, the shape changes of the F cell are impressive and it is surprising that none of the N or E cadherin or integrins have specific enrichment. It would have been a nice add on to hint on potential molecular mechanisms. However, I agree with the reviewers that it is going beyond the scope of this study. Overall, the study is impressive, very complete and can be published as it is in Nature Cell Biology.

Reviewer #3:**Remarks to the Author:**

The authors of this study have appropriately addressed my concerns. No further comments.

Final Decision Letter:

Dear Dr Tapon,

I am pleased to inform you that your manuscript, "Co-option of epidermal cells enables touch sensing", has now been accepted for publication in Nature Cell Biology.

Please note that *Nature Cell Biology* is a Transformative Journal (TJ). Authors may publish their research with us through the traditional subscription access route or make their paper immediately open access through payment of an article-processing charge (APC). Authors will not be required to make a final decision about access to their article until it has been accepted. Find out more about Transformative Journals

If you have not already done so, we strongly recommend that you upload the step-by-step protocols used in this manuscript to the Protocol Exchange (www.nature.com/protocolexchange), an open online resource established by Nature Protocols that allows researchers to share their detailed experimental know-how. All uploaded protocols are made freely available, assigned DOIs for ease of citation and are fully searchable through nature.com. Protocols and Nature Portfolio journal papers in which they are used can be linked to one another, and this link is clearly and prominently visible in the online versions of both papers. Authors who performed the specific experiments can act as primary authors for the Protocol as they will be best placed to share the methodology details, but the Corresponding Author of the present research paper should be included as one of the authors. By uploading your Protocols to Protocol Exchange, you are enabling researchers to more readily reproduce or adapt the methodology you use, as well as increasing the visibility of your protocols and papers. You can also establish a dedicated page to collect your lab Protocols. Further information can be found at www.nature.com/protocolexchange/about

With kind regards,
Daryl

Daryl J.V. David, PhD

Senior Editor, Nature Cell Biology
Nature Portfolio

Heidelberger Platz 3, 14197 Berlin, Germany
Email: daryl.david@nature.com
ORCID: <https://orcid.org/0000-0002-9253-4805>